# SOX2+ sustentacular cells are stem cells of the postnatal adrenal medulla

Alice Santambrogio[1,2,9], Yasmine Kemkem [1,9], Thea L. Willis [1],
Ilona Berger [1,2], Maria Eleni Kastriti [3], Louis Faure [3], John P. Russell [1],
Emily J. Lodge [1], Val Yianni[1], Bence Kövér[1], Rebecca J. Oakey [4],
Barbara Altieri [5], Stefan R. Bornstein[2,6,7], Charlotte Steenblock [2],
Igor Adameyko [3,8] & Cynthia L. Andoniadou [1,2] ✉

Renewal of the catecholamine-secreting chromaffin cell population of the adrenal medulla is necessary for physiological homeostasis throughout life. Definitive evidence for the presence or absence of an adrenomedullary stem cell has been enigmatic. In this work, we demonstrate that a subset of sustentacular cells endowed with a support role, are in fact adrenomedullary stem cells. Through genetic tracing and comprehensive transcriptomic data of the mouse adrenal medulla, we show that cells expressing *Sox2*/SOX2 specialise as a unique postnatal population from embryonic Schwann Cell Precursors and are also present in the normal adult human adrenal medulla. Postnatal SOX2+ cells give rise to chromaffin cells of both the adrenaline and noradrenaline lineages in vivo and in vitro. We reveal that SOX2+ stem cells have a second, paracrine role in maintaining adrenal chromaffin cell homeostasis, where they promote proliferation through paracrine secretion of WNT6. This work identifies SOX2+ cells as a true stem cell for catecholamine-secreting chromaffin cells.

The adrenal medulla is responsible for the body's reaction to acute stress and mediates the 'fight or flight' response through the production and release of the catecholamines adrenaline, noradrenaline and low levels of dopamine, by specialised neuroendocrine chromaffin cells. Catecholamines target multiple organs to help increase oxygenation of muscles, blood pressure and heart output, blood sugar levels, attention and focus and promote vasoconstriction as well as enhance memory performance[1,2]. Diseases of the adrenal, such as congenital adrenal hyperplasia, dopamine beta-hydroxylase deficiency and tumours (pheochromocytomas and the related paragangliomas) lead to disruption in catecholamine regulation with life-threatening consequences[3–5].

The study of adrenal medulla homeostasis, the consequences of homeostatic perturbation, and prospects for regenerative approaches are lacking due to an incomplete characterisation of cell types in this organ. Previous in silico studies of the adrenal gland have delivered insights into adrenocortical cell transcriptome but failed to provide a characterisation of adrenomedullary cell subtypes. This is mostly due to cell isolation protocols being optimised for the adrenal cortex, resulting in very low viable cell numbers from the medulla[6–8]. The existence of a putative adrenomedullary stem cell, capable of giving rise to new chromaffin cells in vivo, has previously been postulated but not identified[9–11]. Instead, divisions in this slow-turnover organ have

[1]Centre for Craniofacial and Regenerative Biology, King's College London, London, UK. [2]Department of Medicine III, Faculty of Medicine Carl Gustav Carus, Technische Universität Dresden, Dresden, Germany. [3]Department of Neuroimmunology, Center for Brain Research, Medical University Vienna, Vienna, Austria. [4]Department of Medical and Molecular Genetics, King's College London, London, UK. [5]Department of Internal Medicine I, Division of Endocrinology and Diabetes, University Hospital, University of Würzburg, Würzburg, Germany. [6]Division of Diabetes and Nutritional Sciences, King's College London, London, UK. [7]Department of Endocrinology, Diabetology and Clinical Nutrition, University Hospital Zurich (USZ) and University of Zurich (UZH), Zurich, Switzerland. [8]Department of Physiology and Pharmacology, Karolinska Institutet, Stockholm, Sweden. [9]These authors contributed equally: Alice Santambrogio, Yasmine Kemkem. ✉e-mail: cynthia.andoniadou@kcl.ac.uk

been attributed to chromaffin cells[12,13]. The poor understanding of the adrenomedullary transcriptomic landscape has even hindered the discovery of specific cell markers within the heterogeneous chromaffin subtypes, such as the noradrenaline-producing chromaffin cells which, lacking a specific marker, are instead identified by the lack of expression of phenylethanolamine N-methyltransferase (PNMT), which catalyses the conversion of noradrenaline to adrenaline[6–8].

In this work, using single-cell RNA sequencing with cell isolation methods optimised for adrenomedullary cells, we identify a postnatal adrenomedullary stem cell population expressing the transcription factor SOX2. Through in vivo lineage tracing and in ovo assays, we demonstrate that these are a specialised long-lived population of stem cells originating and specialising from the neural crest-derived embryonic precursors of the adrenal medulla, termed Schwann cell precursors (SCPs)[14–16]. These adult SOX2$^+$ stem cells contribute to the generation of new noradrenaline- and adrenaline-secreting chromaffin cells throughout life and promote organ proliferation through paracrine signalling. The identification of this adrenomedullary stem cell population holds promise for applications in regenerative medicine in neuroendocrine structures. Pheochromocytomas and paragangliomas, which present some of the highest rates of gene heritability across all tumours, are understudied due to lack of genetic tools. This adrenomedullary stem cell population constitutes an ideal target for oncogenic mutations, in the quest for the generation of novel animal models.

## Results

### Transcriptomic analysis of the postnatal adrenal medulla

To investigate the postnatal cell composition of the adrenal medulla, we performed droplet-based single-cell RNA sequencing on 10 mouse adrenals that were manually dissected to remove the majority of the cortex, at postnatal day (P) 15. At this time of rapid postnatal growth[17], we seek to capture a plastic state that is still representative of normal homeostasis of the organ (schematic Fig. 1A) ($n = 5$ mice, mixed sex). Adrenomedullary cells were subset in silico using markers listed in Supplementary Fig. 1B, and all cortex, endothelial and immune cell types were excluded (Supplementary Fig. 1A, B). Following quality control (Supplementary Fig. 1C, D) unsupervised clustering of 2708 medullary cells revealed 8 distinct transcriptional signatures (Fig. 1B, C). Cluster identity was assigned based on differential expression of known cell markers (Fig. 1D).

All chromaffin cells express tyrosine hydroxylase (TH), which converts tyrosine to L-DOPA, the precursor of dopamine. The action of dopamine-β-hydroxylase, encoded by *Dbh*, converts dopamine into noradrenaline, and subsequently, PNMT converts noradrenaline to adrenaline. Two types of chromaffin cells exist; a first type that expresses PNMT and secretes adrenaline, and a second type, present in the minority, not expressing PNMT and secreting noradrenaline. The differentiated chromaffin cells were further divided into 5 clusters: three consistent with adrenaline-producing signatures (Clusters 0–821 cells, 1–717 cells, 5–105) expressing *Chga*, *Th*, *Ddc*, *Dbh* and *Pnmt*, revealing heterogeneity amongst this population; one noradrenaline-producing (Cluster 2–382 cells) expressing *Chga*, *Th*, *Ddc* and *Dbh* but not *Pnmt*. In the literature, postnatal noradrenaline chromaffin cells were so far only recognised owing to the lack of *Pnmt* expression and were lacking identifying markers. Our study identifies three such unique markers expressed both within clusters 2 as well as the less-committed cluster 3, *Cox8b*, *Lix1* and *Penk* (Supplementary Fig. 1E). *Penk* encodes a preproprotein, whose products have previously been identified in chromaffin cell extracts[18] and its expression confirmed in human fetal chromaffin cells[15]. Immunofluorescence staining for PENK, demonstrates overlap with TH, marking all chromaffin cells, and mutually exclusive expression with PNMT, marking adrenaline-producing chromaffin cells (Supplementary Fig. 1F). We therefore consider PENK a reliable marker of noradrenaline chromaffin cells in

mouse and further confirm its expression in human normal adult adrenal (Supplementary Fig. 1G). Consistent with previous literature[12,13], we additionally identify a fifth cluster of committed chromaffin cells, designated as cycling chromaffin cells (cluster 7 Cycling Chr – 92 cells) expressing chromaffin cell markers (including both *Pnmt* and *Penk*), as well as *Mki67* and *Top2a*. Cell cycle analysis confirmed that the majority of cluster 7 cells were in the G2M phase. Cells in G2M were also identified across all clusters, including sustentacular cells (Fig. 1E).

We detected the presence of a cluster not expressing any chromaffin cell markers, indicative of possible progenitor/stem cells. This was designated as the sustentacular cell cluster, based on the known expression of markers of previously described sustentacular cells[19–22], a signature partly shared by SCPs, the embryonic progenitors of the adrenal medulla. These included markers *Plp1*, *Lgi4*, *Fabp7*, *Sfrp1* and *Cdh19*[14,23] (Cluster 4–170 cells), (Fig. 1F, Supplementary Fig. 1H). The stem/progenitor markers *Sox10*, *S100b*, *Gfap* were expressed among this postnatal population, as well as *Sox2*, not previously reported but a marker of multiple progenitor/stem cells[24–27] (Fig. 1F). These genes all exhibited transcriptional heterogeneity amongst the sustentacular cell population and were additionally expressed, albeit at reduced expression levels, in two additional clusters. These were designated transitioning cell clusters (Clusters 3 T-NorAdrChr – 328 cells and 6 T-AdrChr – 93 cells), as they shared transcriptional signatures with both chromaffin and sustentacular cell markers and are likely committing progenitors of the two types of differentiated chromaffin cells (Fig. 1D, Supplementary Fig. 1I). This observation is supported by pseudotime inference, which predicts chromaffin cells arising from sustentacular cells via the transitioning clusters (Supplementary Fig. 1J). Rare transitioning cells, that express low levels of both sustentacular and chromaffin markers, can be identified in tissue sections (Supplementary Fig. 1K). In summary, these data support the presence of a postnatal adrenomedullary progenitor/stem cell population within the previously termed sustentacular cells and indicate two branches of progenitors during commitment to either noradrenaline- or adrenaline-producing chromaffin cells.

### SCP-derived SOX2$^+$ cells persist throughout life

We next sought to determine if the expression of SOX2 marks a distinct subset of this sustentacular cell population. Immunohistochemistry using antibodies against SOX2 on sections of murine adrenals revealed that SOX2$^+$ cells are present in the adrenal medulla during the early postnatal period and adulthood (Fig. 2A). Quantification of SOX2$^+$ cells as a proportion of the total cells in the medulla reveals an increase in the SOX2$^+$ cell proportion between P15 and P17 (4.95% to 7.04%), a time of high cycling activity, as determined by quantification of Ki-67+ cycling cells (Supplementary Fig. 2A), followed by a gradual decrease until P42 and maintenance of the proportion of SOX2$^+$ cells until P365 (Fig. 2B). We did not observe a difference in SOX2$^+$ cell proportions between sexes (Supplementary Fig. 2A), of relevance since previous reports indicate a discrepancy in the volume of murine medulla based on sex[17]. Separation of sequenced cells by sex (Supplementary Fig. 1D) revealed expected differences, with higher expression of *Xist* and *Tsix* in females, as well as Y-linked *Ddx3y* and *Eif2s3y* in males. Beyond these, only four genes showed significant differences in expression: in females, there is an increase of *Scg2* in adrenaline chromaffin cells and of *Rnd2* in noradrenaline chromaffin cells, and in males there is an increase of two major histocompatibility complex genes *B2m* and *H2-D1* in the transitioning noradrenaline cell cluster (Supplementary Fig. 1D). Immunostaining for SOX2 in human adrenals into advanced age (from patients aged 17, 29, 48, 55, 56, 71), confirms the presence of SOX2$^+$ cells in the human adrenal medulla across both sexes (Supplementary Fig. 2B). To validate overlap with sustentacular markers we used the *Sox2$^{eGFP/+}$* mouse line where EGFP is expressed under the control of *Sox2* regulatory elements[25] and confirmed that all SOX2$^+$ cells

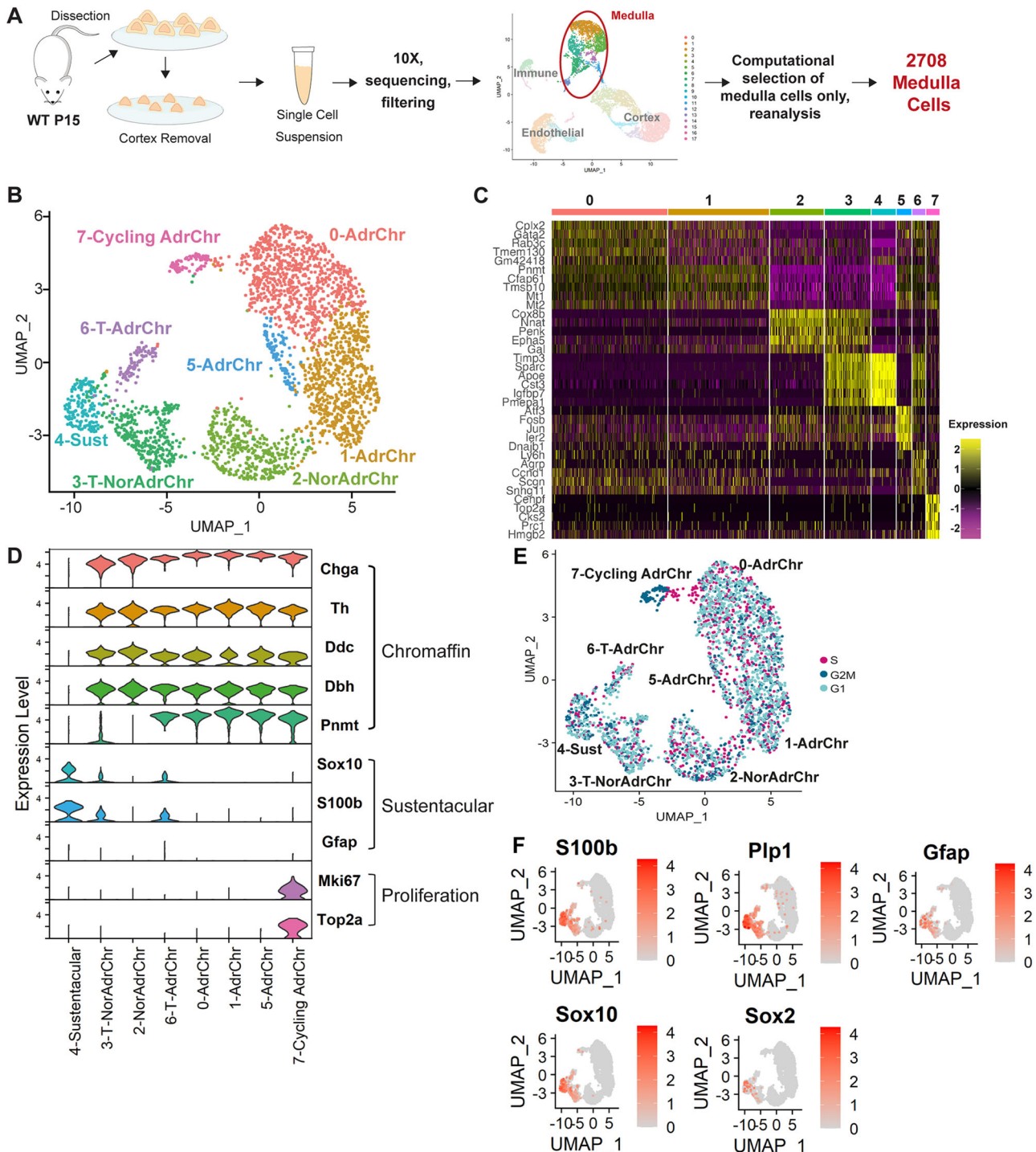

**Fig. 1 | Single-cell RNA sequencing of the mouse adrenal medulla.**
**A** Experimental workflow. **B** UMAP of adrenomedullary cell types (2708 cells).
**C** Heatmap showing the transcriptional signatures of 8 clusters (top 5 differentially expressed genes). **D** Violin plots indicating expression of different markers to identify each cluster. **E** UMAP showing the distribution of cell cycle states over the dataset. Computationally identified percentages of medullary cells in different cell cycle phases: 44.61% (1208 cells) are in G1, 31.75% (859 cells) are in S, 23.67% (641 cells) in G2/M; **F** Featureplots of known sustentacular cell and SCP markers *S100b*, *Plp1*, *Gfap*, *Sox10*, and of newly identified marker *Sox2*. Colour scale represents Log-normalised expression level, with 0 (grey) representing no expression, and 4 (red) representing highest expression.

express EGFP (Supplementary Fig. 2C). Double-immunofluorescence staining confirms that SOX2⁺ cells express classical sustentacular cell markers SOX10, S100B and GFAP (Fig. 2C). RNAscope mRNA in situ hybridisation confirms transcripts of both *Sox2* and *S100b* or *Gfap* in the same cells and additionally reveals overlap of *Sox10* and *Plp1* with *Sox2*, affirming the shared signature with SCPs (Fig. 2D). The overlapping relationship between these five markers in the transcriptomic data, is shown in Supplementary Fig. 2C. Analysis of a published developmental SCP dataset[28] reveals that *Sox2* is expressed among 'multipotent hub' cells, and that *Sox10* expression precedes *Sox2* expression in the chromaffin cell commitment trajectory (Fig. 2E). The *Sox2* regulon is active in the uncommitted state of the chromaffin

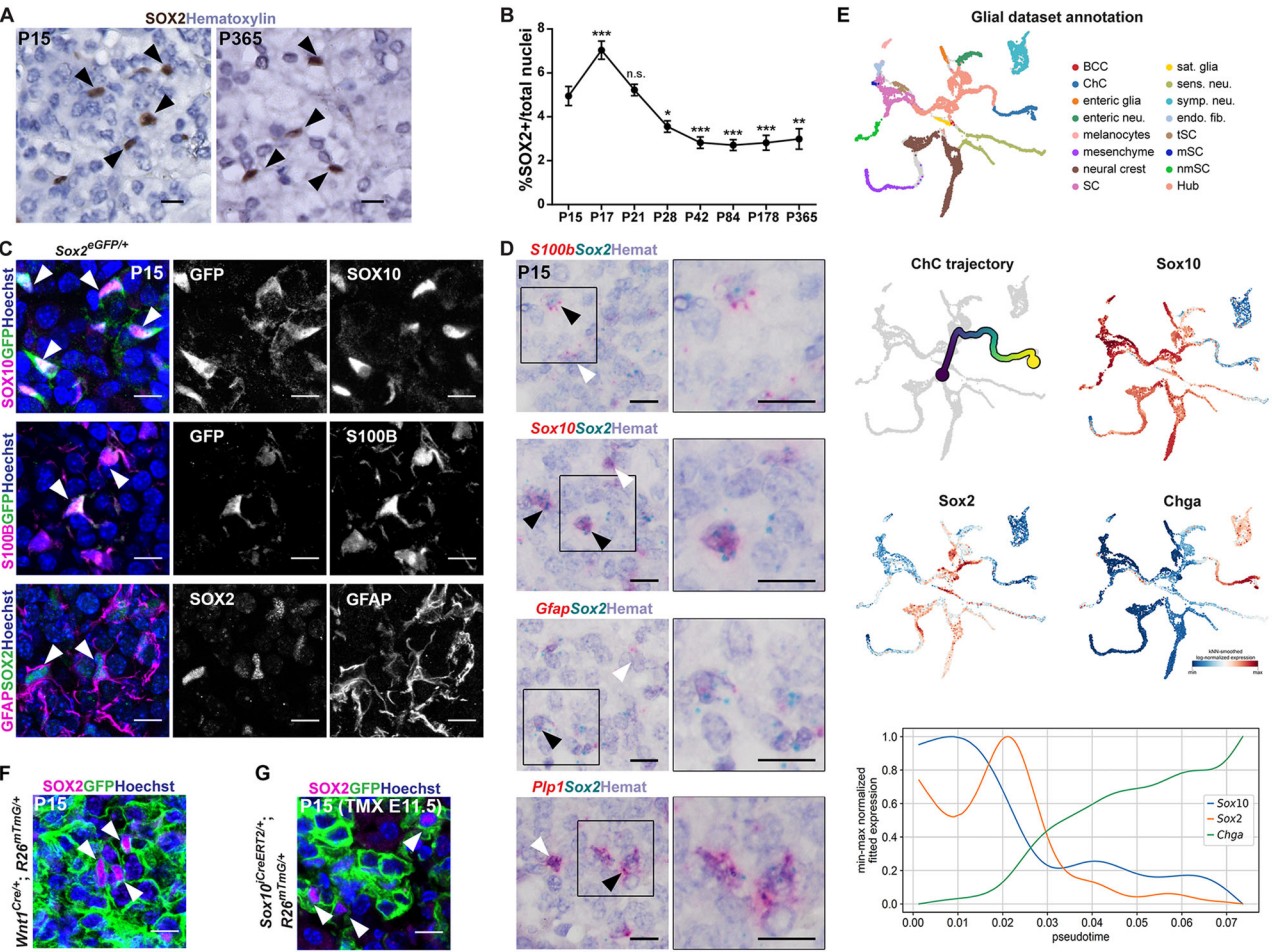

**Fig. 2 | SOX2⁺ cells are present in the adrenal medulla and are derived from Schwann Cell Precursors. A** Immunohistochemistry with antibodies against SOX2 (brown) at P15 and P365 in wild type adrenal medullae. Nuclei counterstained with Hematoxylin, scale bar 20 μm. **B** Quantification of SOX2⁺ cells over the total nuclei of adrenal medulla. $n = 3$ females, 3 males for each timepoint, except for 365-day lineage tracing $n = 2$ females, 2 males, 3 technical replicates for each timepoint, plotted mean and SEM. One-way ANOVA multiple comparisons test: P15 vs. P17 (P-value 0.0008); P15 vs. P21 (P-value 0.9925); P15 vs. P28 (P-value 0.0386); P15 vs. P42 (P-value 0.0006); P15 vs. P84 (P-value 0.0003); P15 vs. P178 (P-value 0.0006); P15 vs. P365 (P-value 0.0013). Source data are provided as a Source Data file. **C** Immunofluorescence staining of P15 *Sox2ᵉᴳᶠᴾ/⁺* adrenal medulla using antibodies against SOX10 (magenta) or S100β (magenta) and GFP (green), shows double-positive cells in both (arrowheads). Immunofluorescence staining of a P15 wild type (WT) sample using antibodies against GFAP (magenta) and SOX2 (green) shows double-positive cells (arrowheads). Nuclei counterstained with Hoechst, scale bars 10 μm. $n = 2$ females, 2 males, 3 technical replicates. **D** RNAscope mRNA in situ hybridisation on wild type P15 samples shows double-positive cells for *S100b* (red) and *Sox2* (blue), *Gfap* (red) and *Sox2* (blue), *Sox10* (red) and *Sox2* (blue), *Plp1* (red) and *Sox2* (blue) respectively - (black arrowheads) or single positive (white arrowheads); all nuclei counterstained with Hematoxylin, scale bar 10 μm. $n = 2$ females, 2

males, 2 technical replicates. **E** UMAP of the neural crest and SCP lineages between 9.5dpc and 12.5dpc from ref. 28: BCC, boundary cap cells; ChC, chromaffin cells; enteric neu., enteric neurons; SC, Schwann cells; sat. glia, satellite glia; sens. neu., sensory neurons; symp. neu., sympathetic neurons, endo. fib., endoneurial fibroblasts; tSC, terminal Schwann cells; mSC, myelinating Schwann cells; nmSC, non-myelinating Schwann cells. Trajectory of chromaffin cell (ChC) transitioning from the Hub cells (range from blue (uncommitted hub cell) to yellow (chromaffin cell)). Featureplots showing expression of *Sox10*, *Sox2*, *Chga* (scale Log-normalised expression with *k*-nearest neighbour (kNN) smoothing, ranging from blue (low expression) to red (high expression)). Graph of single features for *Sox10* (blue), *Sox2* (orange) and *Chga* (green) along pseudotime. *Sox10* expression precedes that of *Sox2* in this trajectory, which is maintained until expression of *Chga* marking chromaffin cells. **F** Immunofluorescence on P15 mouse adrenals from *Wnt1Cre/⁺;R26ᵐᵀᵐᴳ/⁺* genotypes. Immunostaining with antibodies against SOX2 (magenta) and GFP (green) shows double-positive cells (arrowheads). Nuclei are counterstained with Hoechst, scale bars 10μm. $n = 1$ male, 1 female, 2 technical replicates. **G** Immunofluorescence on P15 mouse adrenals from a *Sox10CreERT2/⁺;R26ᵐᵀᵐᴳ/⁺* line induced with tamoxifen (TMX) at 11.5dpc – immunostaining with antibodies against SOX2 (magenta) and GFP (green) shows double-positive cells (arrowheads). Nuclei counterstained with Hoechst, scale bars 10 μm.

commitment trajectory (Supplementary Fig. 2D), and SCENIC analysis reveals that the majority of *Sox2* and *Sox10* targets are distinct (*Sox10*-only 447, *Sox2*-only 131, shared 38) (Supplementary Fig. 2D, Supplementary Data 1). The top 10 markers correlated with *Sox2* expression across all cells include genes highly expressed in postnatal sustentacular cells (*Fabp7, Sparc, Zfp36l1, Serpine2 Sox10*), and the Hippo pathway regulator *Wwtr1* (Supplementary Fig. 2E, F). Anti-correlated genes (analysis only among *Sox2*-expressing cells) include chromaffin cell markers *Chga, Chgb* and *Th* (Supplementary Fig. 2G), supporting the notion that *Sox2* expression needs to be downregulated for acquisition of a chromaffin cell state. To determine if these SOX2⁺

sustentacular cells are indeed derived from SCPs in the embryo, we carried out lineage tracing of embryonic SCPs. Using *Wnt1Cre/⁺;R26ᵐᵀᵐᴳ/⁺* we labelled the neural crest from its specification and using *Sox10ⁱᶜʳᵉᴱᴿᵀ²/⁺;R26ᵐᵀᵐᴳ/⁺* we induced SCPs at 11.5dpc, at a time when they begin to migrate towards the dorsal aorta, subsequently giving rise to the adrenal medulla. Immunofluorescence staining using GFP and SOX2 antibodies, reveals that descendants of *Wnt1* and *Sox10* expressing cells (GFP+) include the entire SOX2⁺ population at P15. Specifically, 100% of SOX2⁺ cells are GFP⁺ in *Wnt1Cre/⁺;R26ᵐᵀᵐᴳ/⁺* adrenals, and 99.58% in *Sox10ⁱᶜʳᵉᴱᴿᵀ²/⁺;R26ᵐᵀᵐᴳ/⁺* adrenals, where recombination following tamoxifen induction can be incomplete (Fig. 2F, G, $n = 3$ per genotype).

In order to determine if postnatal SOX2[+] cells are a distinct specialised cell type from SCPs, we compared the signatures of SCPs and postnatal SOX2[+] cells. Isolation of a flow-purified population enriched for SOX2[+] cells using the *Sox2*[eGFP/+] mouse line, allowed single-cell RNA sequencing of 1563 cells, 507 of which are expressing high levels of *Sox2* (Supplementary Fig. 3A). Using the CONOS package, the cell identities from the developmental glia dataset were aligned to the postnatal dataset. This shows that the SOX2[+] sustentacular cells are dissimilar to SCPs (Supplementary Fig. 3B). The population to which they exhibit the highest similarity is postnatal non-myelinating Schwann cells. The top 50 differentially expressed genes from SOX2[+] sustentacular cells are scored on both datasets (Supplementary Fig. 3C). SCENIC analysis comparing *Sox2* regulons, reveals little overlap between *Sox2* targets in developmental glia with cells of the postnatal medulla (121 unique *Sox2* targets in developmental glia, 162 *Sox2* targets in the postnatal medulla, 7 shared targets) (Supplementary Fig. 3D, Supplementary Data 2). In summary, SCP-derived *Sox2*-expressing cells of the postnatal adrenal medulla are a distinct uncommitted postnatal population.

### SOX2[+] cells of the adrenal medulla are stem cells ex vivo

To determine if *Sox2*-expressing cells have the potential for self-renewal, we cultured dissociated adrenomedullary cells in stem-cell promoting media under adherent conditions. Flow sorting of EGFP[+] and EGFP[-] cells from *Sox2*[eGFP/+] mice at P15 and plating in clonogenic conditions, showed that EGFP[+] (SOX2[+]) cells only, can give rise to colonies, which can be passaged (Fig. 3A, B, Supplementary Fig. 4A–C). These colonies contained an expanded population of SOX2[+] cells as revealed by immunofluorescence staining and by flow cytometry for EGFP expression (Fig. 3C, D). These data combined, render *Sox2*-expressing cells as a putative postnatal stem cell population. To establish if SOX2[+] cells alone are sufficient to give rise to chromaffin cells, we took advantage of a well-established in vivo xenograft culture technique, chorioallantoic membrane (CAM) culture, in chicken embryos[29]. This allows culture of three-dimensional vascularised tissues in an in vivo environment, enabling long-term maintenance. We first used our newly-established in vitro culture system to isolate and expand postnatal SOX2[+] stem cells over 8 days, at which point they are mostly uncommitted, as revealed by EGFP detection by flow sorting where negative cells average 0.48% of the population (Fig. 3D). Purified SOX2[+] cell suspensions (800,000 cells, purified and expanded from 4 to 6 animals) were grafted onto the embryonic CAM (Fig. 3E). Collection of the xenografts 10 days later, revealed that SOX2[+] cells can give rise to compact three-dimensional tissues (*n* = 4 out of 10 CAM assays, Fig. 3F). Endogenous expression of EGFP was detectable in the grafts at collection, suggesting the presence of SOX2[+]cells (Fig. 3F). Immunofluorescence staining using antibodies against chromaffin cell markers TH (8.51% of all cells) and PNMT (adrenaline-expressing chromaffin cells) (5.81% of all cells), confirms that grafts contain differentiated chromaffin cells (*n* = 3 grafts, Fig. 3G). RNAscope mRNA in situ hybridisation using probes against mouse-specific *Th* and mouse-specific *Sox2*, confirms that chromaffin cells are derived from the murine graft and that there is persistence of a *Sox2*-expressing uncommitted population (Fig. 3G).

### SOX2[+] cells of the adrenal medulla are stem cells in vivo

To establish if SOX2[+] cells function as stem cells during homeostasis in vivo, we labelled and lineage-traced *Sox2*-expressing cells in the postnatal adrenal. Tamoxifen induction of *Sox2*[CreERT2/+]; *R26*[mTmG/+] animals was carried out by single injection at P14 and adrenals collected after 72 h (P17), 7 days (P21), 14 days (P28), 28 days (P42), 70 days (P84), 178 days (P192) and 365 days (P379). (Fig. 4A). No-Cre controls or injection with corn oil instead of tamoxifen, did not result in GFP expression (Supplementary Fig. 4D). Lineage tracing over 365 days reveals an expansion of GFP[+] clones in the adrenal medulla (Fig. 4B). Quantification of GFP[+] as a proportion of the total nuclei demonstrates

an increase over time: initial labelling of 3.4% GFP[+] cells at 72 h post induction to 24.8% at 1 year post induction (Fig. 4C), at which point a proportion of GFP[+] cells (8.46%) are also SOX2[+] (Fig. 4D), indicating maintenance of the stem cell pool. Double-immunofluorescence staining at 178 days with antibodies against GFP and general chromaffin marker TH, adrenaline-specific chromaffin marker PNMT and the newly identified noradrenaline-specific chromaffin marker PENK, confirms GFP[+] cells double-stained with either marker (12.5% TH[+];GFP[+], 9.2% PNMT[+];GFP[+], 2.8% PENK[+];GFP[+]), confirming the derivation of both adrenaline- and noradrenaline-producing chromaffin cells from SOX2[+] sustentacular progenitors (Fig. 4E). Lineage tracing in adult mice induced at P189 (27 weeks, *n* = 4) and analysed 28 days later (P217), reveals that the in vivo potential of *Sox2*-expressing cells to generate chromaffin cells is retained in later life (Fig. 4F). Taken together, our data therefore demonstrate that SOX2[+] adrenomedullary cells are bona fide stem cells.

### WNT ligands from SOX2[+] stem cells promote endocrine cell expansion

We previously reported that SOX2[+] stem cells of a different endocrine organ, the anterior pituitary gland, are instrumental to promote postnatal organ proliferation in a paracrine manner, through the secretion of WNT ligands[30]. To determine if SOX2[+] stem cells of the adrenal medulla share this non-classical stem cell contribution to organ turnover, we mined our single-cell RNA sequencing dataset of the mouse adrenal medulla to first explore the cell types that upregulate the WNT pathway. We found that WNT pathway targets *Lef1* and *Axin2* are both expressed in chromaffin cells, with a bias for the noradrenaline (*Lef1*) and adrenaline (*Axin2*) lineages. *Lgr5*, a WNT pathway potentiator and target is strongly expressed in all committed chromaffin cell clusters. All three targets are expressed in the cycling chromaffin cell cluster (cluster 7) (Fig. 5A). Upregulation of canonical WNT signalling in chromaffin cells was confirmed through immunofluorescence against TH on the TCF/Lef:H2B-GFP reporter line, showing activation of GFP (WNT-responding cells) among TH[+] chromaffin cells (Fig. 5B). Expression of all three WNT targets was absent from the sustentacular/stem cell cluster. To determine the source of WNT ligands we queried expression of all mouse *Wnt* genes. *Wnt1, 2, 2b, 3a, 5b, 7a, 7b, 8a, 8b, 9a, 9b, 10a, 10b,* and *11* were not expressed in any adrenomedullary cell population. Low expression of *Wnt3, 4, 5a* and *16* was detected in sporadic cells across different clusters (Supplementary Fig. 5A). Expression of *Wnt6* was strong in the sustentacular/stem cell cluster, and detectable but weak in the two transitioning noradrenaline and adrenaline clusters (Fig. 5C and Supplementary Fig. 5A). Expression of *Wnt* genes within the isolated SOX2-EGFP[+] cells confirmed robust expression of *Wnt6* as the sole *Wnt* gene (Fig. 5C and Supplementary Fig. 5B). Double mRNA in situ hybridisation using probes against *Wnt6* and *Sox2*, confirms specific expression of *Wnt6* in this stem cell population (Fig. 5D, *n* = 3). WLS (GPR177) is a glycoprotein receptor necessary for WNT secretion. *Wls* expression was detected across all populations of the adrenal medulla including the SOX2[+] cells (Fig. 5E). We specifically deleted *Wls* in SOX2[+] cells (*Sox2*[CreERT2/+];*Wls*[fl/fl]) in a tamoxifen-inducible manner. Mice were induced at P13/14/15 and adrenals collected at P21. Immunofluorescence staining using antibodies against Ki-67 to mark cycling cells, revealed a reduction in overall proliferation in adrenal medullae deficient in *Wls* expression (lack of WNT secretion) from SOX2[+] cells (Fig. 5F, *n* = 6–8 per genotype). Cycling cells are normally found across both *Sox2*-expressing as well as *Th*-expressing cell populations (Fig. 1E, Supplementary Fig. 5C). Immunofluorescence staining using antibodies against TH and Ki-67, shows a drop in the percentage of cycling cells across both chromaffin (TH[+]) and non-chromaffin (TH[-]) cells in the adrenal medulla of *Sox2*[CreERT2/+];*Wls*[fl/fl] mutants (Fig. 5G, *n* = 3 per genotype). Similarly, *Sox2*[CreERT2/+];*Wls*[fl/fl] mutants display a reduced incorporation of EdU in TH[+] chromaffin cells after a 2 h pulse, indicating a lower proportion of cells in S-phase (Supplementary Fig. 5D, *n* = 3 per genotype). These results confirm that

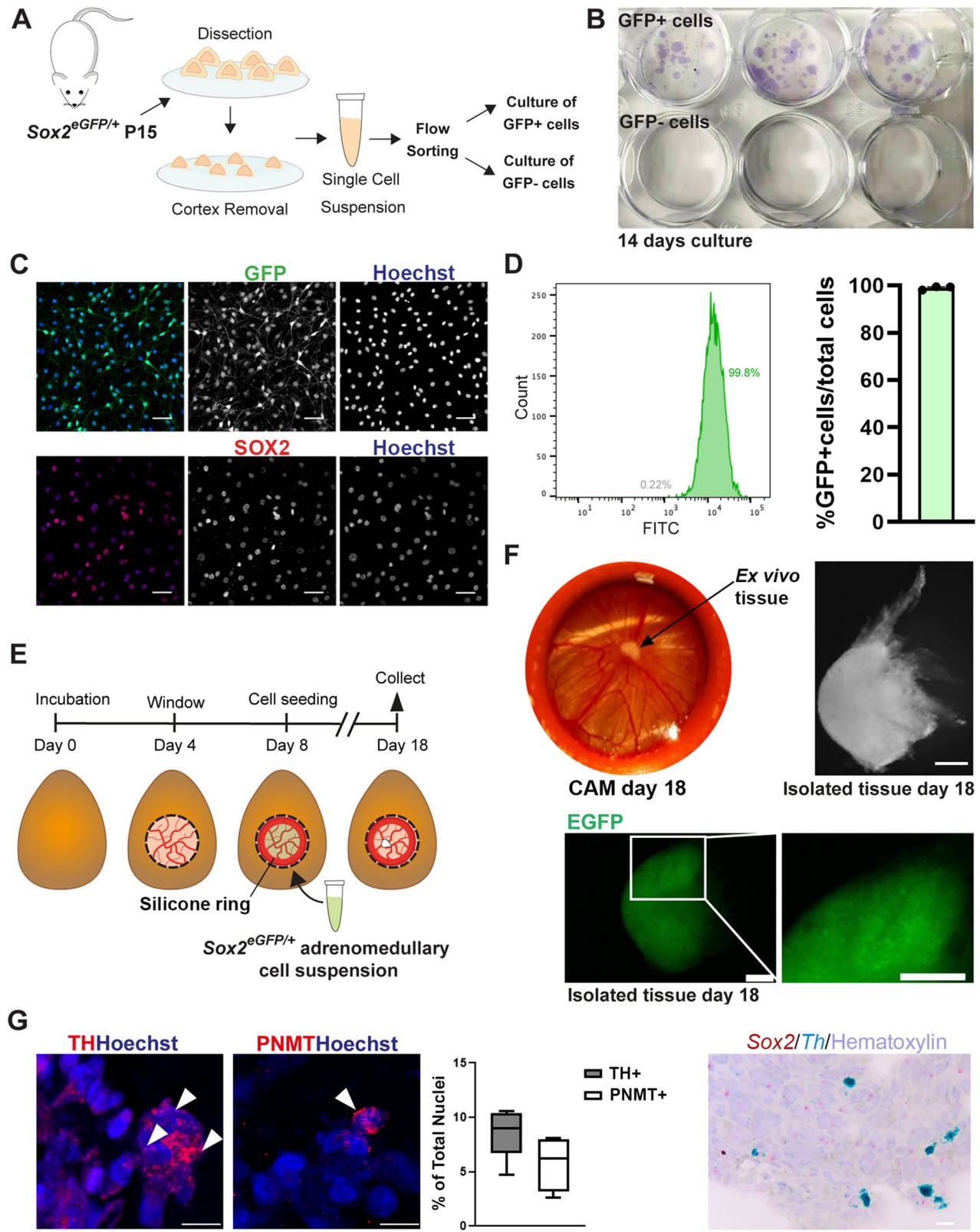

SOX2$^+$ cells promote proliferation in the adrenal medulla, through paracrine secretion of WNT6.

## Discussion

Here, we reveal the existence of postnatal adrenomedullary stem cells, which give rise to new chromaffin cells of both the adrenaline and the noradrenaline lineages throughout life, as well as contribute to the normal turnover of chromaffin cells through paracrine signalling. Employing in vivo studies in mouse, we confirm that this specialised *Sox2*-expressing somatic stem cell population derives from *Sox10*-expressing embryonic SCPs of the neural crest and becomes a stem cell population entirely distinct from SCPs. Comprehensive single-cell transcriptome analyses of the murine adrenal medulla were lacking from the literature, since methodologies to dissociate adrenal tissue

**Fig. 3 | Adrenomedullary SOX2⁺ cells have stem cell properties in vitro and in ovo. A** Experimental workflow. **B** Crystal violet staining of fixed cell colonies following 14-day culture of GFP⁺ (SOX2⁺) and GFP⁻ (SOX2⁻) *Sox2^eGFP/+* cells under clonogenic conditions. **C** Immunofluorescence staining of GFP⁺ primary cells from *Sox2^eGFP/+* medulla cultured for 14 days: GFP (green), SOX2 (red), nuclei stained with Hoechst, scale bar 50μm. *n* = minimum 8 adrenals pooled from minimum 4 mixed-sex mice, 4 technical replicates. **D** Quantification of GFP⁺ cells via flow cytometry after 14 days of culture, bar graph *n* = 3 independent biological replicates. **E** Experimental design of chick chorioallantoic membrane (CAM) assays for ex vivo 3D xenograft culture. **F** Representative images of resulting xenograft before removal from CAM (left) and after isolation (right) following 18 days of incubation.

Representative images of wholemount native EGFP expression in *Sox2^eGFP/+*-derived xenograft (bottom). Scale bars 200 μm. *n* = minimum 8 adrenals pooled from minimum 4 mixed-sex mice, 4 technical replicates. **G** Immunofluorescence staining using antibodies against TH (red) or PNMT (red) on xenografts at day 18 (*n* = 2). Nuclei counterstained with Hoechst. Box plot showing quantification of TH and PNMT positive cells after immunofluorescence staining, as a proportion of total nuclei. Whiskers min to max, the box extends from the 25th to the 75th percentile, line is median. Source data are provided as a Source Data file. RNAscope mRNA in situ hybridisation using mouse-specific probes against *Sox2* (red) and *Th* (blue) on xenografts. Nuclei counterstained with hematoxylin. Scale bars 10 μm.

favour the cortex, with low medullary cell survival. These reveal the molecular features of adrenomedullary stem cells, and clearly identify them as the cells of origin of noradrenaline- and adrenaline-expressing chromaffin subtypes, with distinct transitioning progenitors. Genetic lineage tracing of *Sox2*-expressing cells, confirms the generation of new chromaffin cells of both types throughout postnatal life. The transcriptomic datasets presented can be exploited further by the community, and as proof-of-concept, we present their use for identification of noradrenaline-specific markers, where previously noradrenaline-secreting chromaffin cells were identified only through their lack of marker expression. Validation of PENK as a marker of noradrenaline cells is demonstrated, and these new markers can be proven useful in human pathology, the study of noradrenaline cells and expansion of the genetic toolkit of mouse models in adrenal research (e.g. *Penk-Cre* mice)[31].

Previously, it was unknown if an adrenomedullary stem cell population exists or if new chromaffin cells are only generated from self-duplication[22]. In this study, we not only demonstrate the potential of SOX2⁺ cells in vivo, but show that they can be cultured and expanded in vitro and generate tissue containing neuroendocrine cells when explanted, here illustrated using an in ovo system. The culture systems we established can be further exploited for stem cell-based regenerative approaches in relation to disorders implicating the adrenal medulla e.g. adrenal hypoplasia or dopamine β-hydroxylase deficiency.

In addition to the classical stem cell paradigm, our in vivo results reveal that SOX2⁺ cells can promote turnover in a second way, through the secretion of paracrine ligands, and we identify WNT6 as a key ligand in this process. The pattern of WNT target activation suggests that WNT6 from the stem cells primarily acts in a cell non-autonomous (paracrine) fashion, rather than cell-autonomous (autocrine). In an organ where a main source of proliferation is the committed neuroendocrine cells, paracrine signalling from stem cells can safeguard a robust response to changing physiological demand, without depleting the stem cell pool. The paradigm of stem cells acting as signalling hubs to regulate proliferation of their neighbouring descendants was previously demonstrated in pituitary gland stem cells[25], and further shown to underlie tumour formation[24]. It supports the possibility that adrenomedullary stem cells may contribute cell autonomously and cell non-autonomously to adrenal tumour pathogenesis. Since an established stem cell population had not been identified, the current dogma dictates the cell-of-origin of pheochromocytoma and the related paraganglioma tumours as being specialised neuroendocrine cells[32]. It can be postulated that the adrenomedullary stem cells identified and characterised in this study may be involved in the initiation or progression of tumours, and our findings can support study of these processes and the generation of disease models.

## Methods

### Ethical considerations
Studies using human adrenals were carried out under King's College London ethical approval with KCL Ethics Reference LRS-19/20-20118, as part of the adrenal tumour registry project of the European Network

for Adrenal Tumours ENS@T (European Network for the Study of Adrenal Tumours). Informed consent was obtained for all samples, patients were not compensated for participation. All animal studies were performed under compliance of the Animals (Scientific Procedures) Act 1986, Home Office Licences P5F0A1579 (mouse) and P8D5E2773 (chicken), as well as KCL Biological Safety approval for project 'Function and Regulation of Adrenal Stem Cells in Mammals'.

### Animals
Procedures were carried out in compliance with the Animals (Scientific Procedures) Act 1986, Home Office licence and King's College London ethical review approval. All mouse colonies were maintained under 12:12 h light/dark cycle and fed *ad libitum*. All mouse lines used were previously published: *Sox2^eGFP/+*[25], *Sox2^CreERT2/+*[24], *Wnt1^Cre/+*[33], *Sox10^iCreERT2/+*[34], *R26^mTmG/+*[35], *Wls^fl/fl*[36]. All mice were bred and maintained on mixed backgrounds and consistently backcrossed on CD1. For Cre recombination, Tamoxifen (Sigma, T5648) was injected intraperitoneally with a single dose of 0.15 mg/g body weight in postnatal mice, except in *Wls* experiments where single dose injections were given on three consecutive days. Pregnant females were injected by a single intraperitoneal injection of tamoxifen, capped at 1.5 mg and one dose of Progesterone (Sigma P0130) at 0.75 mg. For pulse EdU incorporation, EdU was injected by single intraperitoneal injection at a concentration of 50 mg/kg body weight, 2 h before sample collection. Mice were killed humanely via Schedule 1 approved methods: cervical dislocation followed by resection of the femoral artery or $CO_2$ exposure followed by resection of the femoral artery.

### Human samples
Normal adrenal human tissue samples were obtained from the University Hospital Würzburg (Germany). Normal adrenal glands removed as part of tumour nephrectomy and proven to be histologically normal, showing no neoplastic tissue.

### Fluorescent activated cell sorting
Adrenal glands from *Sox2^eGFP/+* mice were dissected and tissue was dissociated as described for primary cell culture. At the last step, cells were resuspended in FACS buffer (2.5% HEPES solution 1 M (Sigma), 1% FBS in PBS), passed through a 40 μm cell strainer (Corning, CLS431750) and stained with DAPI 0.05 μg/ml (Biolegend, 422801), before being flow sorted by a FACSAria Cell Sorter (BD Biosciences). Adrenal medullae from wild-type littermates were used as a negative control.

### 10x Single-cell RNA sequencing and computational analysis - postnatal adrenal medulla
10 adrenals from 5 mixed sex P15 mice were dissected on ice, surrounding fat and excess adrenal cortex were removed manually, including innervating nerves of the cortex. Medullae were placed in an enzymatic digestion mix containing 50 μg/ml DNAse I (Sigma, D5025), 10 mg/ml Collagenase II (Worthington, LS004177), 2.5 μg/ml Fungizone (Gibco, 15290026), 0.1X Trypsin-EDTA (Sigma, 59418C) in 1X

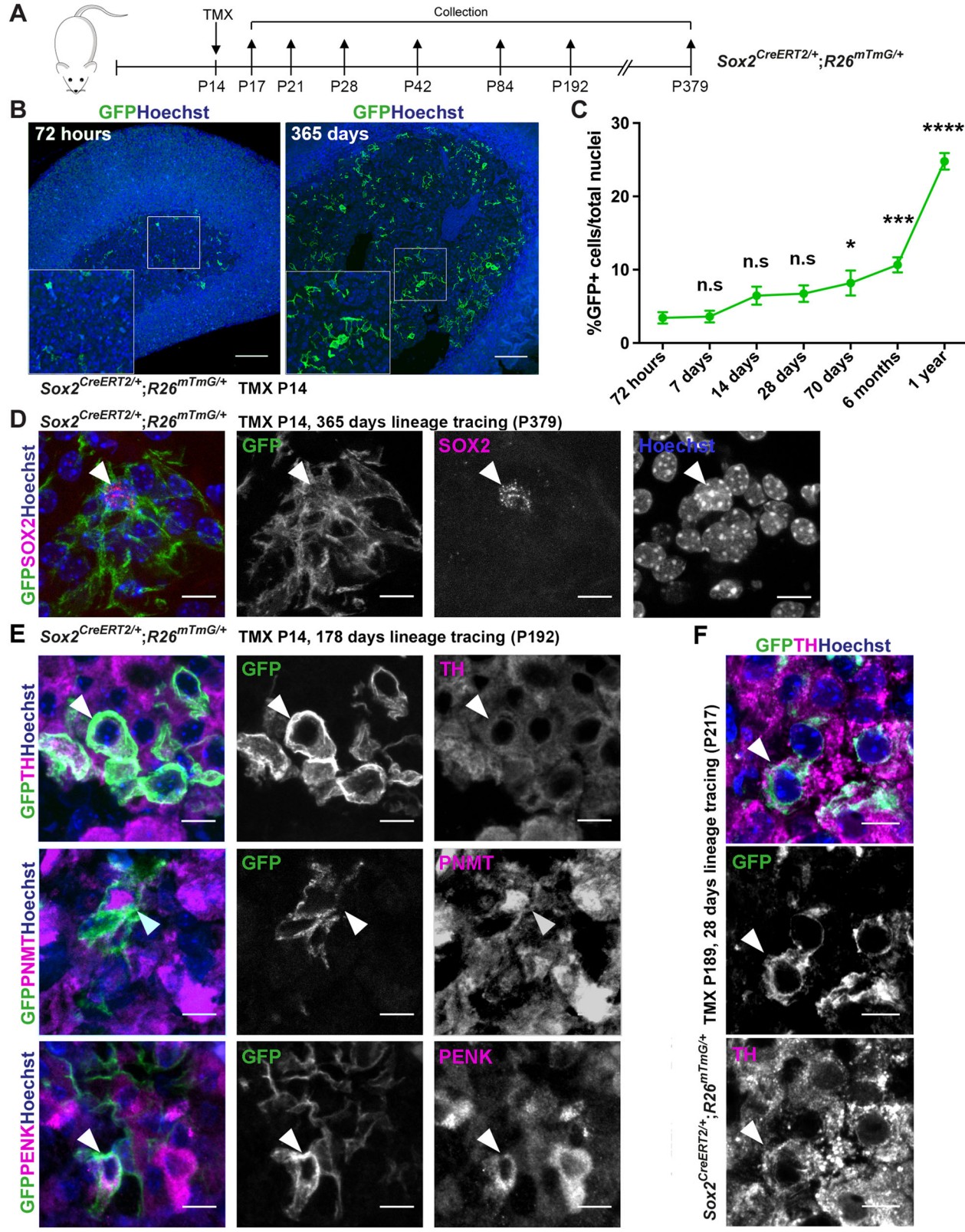

Hank's Balanced Salt Solution (HBSS) (Gibco, 14025050) and incubated at 37 °C for 15 min. Enzymes were inactivated by addition of 10 times volume of serum-containing Base Media: DMEM/F-12 (Gibco, 31330-038) + 5% FBS (Merk, F0804) + 50 μ/ml Penicillin-Streptomycin (Gibco, 15070063). The cell suspension was centrifuged at $281 \times g$ for 5 min at room temperature, washed twice in PBS and pellets were resuspended in a solution of HBSS 2.5% FBS. An aliquot of 10,000

viable cells were used for the experiment. Library preparation and sequencing were performed by the BRC Genomics Core at KCL. Library preparation was done using the Chromium 10X Single-cell 3' Reagent Kit v3.1 (10x Genomics, PN-1000121) and a Chromium Controller (10x Genomics) following the manufacturer's protocol. Once obtained, barcoded transcripts from single cells were sequenced with an Illumina HiSeq 2500.

**Fig. 4 | Adrenomedullary SOX2⁺ cells are stem cells in vivo. A** Experimental design indicating tamoxifen (TMX) induction at P14 and timepoints of collection and analysis. **B** Immunofluorescence using antibodies against GFP, on *Sox2^CreERT2/+*; *R26^mTmG/+* adrenals induced with tamoxifen at P14 and collected after 72 h or 365 days. GFP in green, nuclei counterstained with Hoechst. Inserts magnified boxed regions. Scale bar 100 μm. **C** Quantification of GFP+ cells/total nuclei of adrenal medulla at different timepoints. *n* = 3 females, 3 males, for each timepoint, except for 365-day lineage tracing *n* = 2 females, 2 males, 3 technical replicates for each timepoint, plotted mean and SEM. One-way ANOVA multiple comparisons test: 72 h vs. 7 days (*P*-value > 0.9999); 72 h vs. 14 days (*P*-value = 0.2698); 72 h vs. 28 days (*P*-value = 0.2035); 72 h vs. 70 days (*P*-value = 0.0285); 72 h vs. 178 days (*P*-value = 0.0005), 72h vs. 365 days (*P*-value < 0.0001). Source data are provided as a Source Data file. **D** Double immunofluorescence on *Sox2^CreERT2/+*;*R26^mTmG/+* adrenals

induced at P14 and collected after 365 days, using antibodies against GFP (green) and SOX2 (magenta). Arrowhead indicates a double-labelled cell. Nuclei counterstained with Hoechst (blue). Scale bar 10 μm. *n* = 2 females, 2 males, 3 technical replicates. **E** Double immunofluorescence on *Sox2^CreERT2/+*;*R26^mTmG/+* adrenals induced at P14 and collected after 178 days, using antibodies against GFP (green) and specific cell markers (magenta) TH (all chromaffin cells), PNMT (adrenaline chromaffin cells) or PENK (noradrenaline chromaffin cells). Note the presence of double-labelled cells (arrowheads). Nuclei counterstained with Hoechst (blue), scale bar 10μm. *n* = 3 females, 3 males, 3 technical replicates. **F** Double immunofluorescence on *Sox2^CreERT2/+*;*R26^mTmG/+* mice induced at P189 (6 months) and collected after 28 days, using antibodies against GFP (green) and TH (magenta). Note the presence of double-labelled cells (arrowheads). Nuclei counterstained with Hoechst (blue), scale bar 10 μm. *n* = 3 females, 2 males, 6 technical replicates.

Pre-processing of the sequencing datasets was performed by the BRC Genomics Core at KCL using Cell Ranger-4.0.0. Once feature-barcode matrices were obtained, analysis was performed in RStudio with the Seurat package, v3 and 4[37,38], following author instructions. The dataset was subset to exclude cells with <500 or >5000 genes or with >20% mitochondrial counts. After normalisation, the 2000 most variable features were identified, the dataset was scaled and PCA was performed based on the previously identified variable features. Unsupervised clustering using the Louvain algorithm with resolution 0.4 and generation of UMAP was done using the top 10 PCs using the Seurat implementation. Cluster identities were assigned based on markers from the literature, gene counts from clusters showing medulla-specific markers were extracted and the matrix re-analysed with the same parameters. Markers were identified using the Wilcoxon rank-sum test as implemented in Seurat, using a log2FC threshold of 0.25. Cell cycle analysis was performed using the CellCycleScoring function in Seurat, following author specifications. Pseudotime analysis was performed using Monocle 3 (version 1.3.4)[39], following author instructions.

### 10x Single-cell RNA sequencing and computational analysis - postnatal SOX2-EGFP⁺ cells

Thirty adrenals from 15 mixed sex P15 *Sox2^eGFP/+* mice were dissected on ice, dissociated, and GFP+ cells were isolated via FACS, centrifuged at $300 \times g$ for 5 min and resuspended in a solution of HBSS 2.5% FBS. 2000 viable cells were used for the experiment. Droplet-based single-cell RNA sequencing was performed using the Chromium 10X Single Cell 3' Reagent Kit v3 (10x Genomics) and a Chromium Controller (10x Genomics) following the manufacturer's protocol. Cells with <1000 or >5000 genes or with >20% mitochondrial counts were excluded. After normalisation, the 2000 most variable features were identified, the dataset was scaled and PCA was performed based on the previously identified variable features. Unsupervised clustering using the Louvain algorithm with resolution 0.4 and generation of UMAP was done using the top 10 PCs using the Seurat implementation. Markers were identified using the Wilcoxon rank-sum test as implemented in Seurat, using a log2FC threshold of 0.25. To further select only *Sox2* expressing cells, the WhichCell function was used to select only cells with *Sox2* raw counts >0. Once extracting the raw counts from these cells, the dataset was reanalysed with the same parameters.

### Differential expression analysis of SCPs vs *Sox2*-expressing postnatal cells

SCPs were isolated from a 13.5dpc dataset[15] using the parameters described in the paper. This dataset was combined with the *Sox2* expressing cells dataset using Seurat RPCA integration on the top 30 PCs constructed from the 2000 most variable features. Differential expression analysis between SCPs and postnatal *Sox2* expressing cells was performed following Seurat guidelines. The CONOS package[40] was used to align the postnatal dataset to the developmental glial data. Top 50 DE genes from *Sox2* expressing sustentacular cluster 4 were extracted and scored on both datasets.

### Correlation analyses of *Sox2* regulons

Data published by Kastriti et al.[28] were utilised. The SCENIC package[41] was used to extract the regulon activity matrix as well as identified target genes of our postnatal dataset. The Spearman correlation between the activity score of the *Sox2*(+) regulon and the log10 expression of gene transcripts in all processed cells was calculated. Expression values of top correlated and anti-correlated genes were also represented by fitting these with Generalized-Additive model, using the pseudotime assignments of the cells of the trajectory from late NCC/SCP to ChC. Pseudotime and trajectory representation and analysis were carried out using scFates 1.0.8 package[42].

### Tissue processing

For paraffin-embedding, adrenal glands were dissected, surrounding fat was removed and samples were fixed in 10% neutral buffered formalin (NBF) (Sigma, HT501128) overnight at room temperature. Grafts collected from the CAMs were dissected and fixed following the same protocol. The next day, tissue was washed and dehydrated through graded ethanol series and paraffin-embedded. Samples were sectioned at 5μm thickness. For cryo-embedding, adrenal glands were dissected, surrounding fat removed and samples fixed in 4% PFA at 4 °C for 4 h. Adrenals were washed and cryoprotected in 30% Sucrose overnight at 4 °C. Adrenals were embedded in Optical Cutting Temperature compound (VWR, 361603E) and flash-frozen. Samples were cryo-sectioned at 8–12 μm thickness.

### Immunofluorescence and immunohistochemistry on paraffin sections

Paraffin sections were deparaffinised and rehydrated with ethanol series. Antigen retrieval was performed in a Decloaking Chamber NXGEN (Menarini Diagnostics, DC2012-220V) at 110 °C for 3 min using Declere, pH 6.0 (Cell Marque, 921P-04) for immunohistochemistry or Dako Target Retrieval Solution, pH 9.0 (Agilent, S236784-2) for immunofluorescence.

For immunohistochemistry, ImmPRESS Excel Amplified HRP Polymer Staining Kit Anti-Rabbit IgG (Vector Laboratories, MP-7602-50) was used following the manufacturer's instructions. Primary antibodies were used at the concentrations listed in the Supplementary Table 1. Nuclei were stained with Vector Hematoxylin QS (Vector Laboratories, H-3404-100) and slides were mounted in VectaMount Permanent Mounting Medium (Vector Laboratories, H-5000-60).

For immunofluorescence, sections were blocked for 1 h at room temperature in Blocking Buffer (0.15% glycine, 2 mg/ml BSA, 0.1% Triton X-100 in PBS) with 10% sheep serum. Primary antibodies were diluted in Blocking Buffer with 1% sheep serum at the concentrations described in Supplementary Table 1 and incubated overnight at 4 °C. After washing 3 times with PBST (PBS + 0.1% Triton X-100), samples were incubated for 1 h at room temperature in secondary fluorophore-conjugated antibodies (dilution 1:500, listed in Supplementary Table 1) and Hoechst (Life Technologies, H3570) (dilution 1:10,000) in blocking buffer with 1% serum. Tyrosine Hydroxylase and

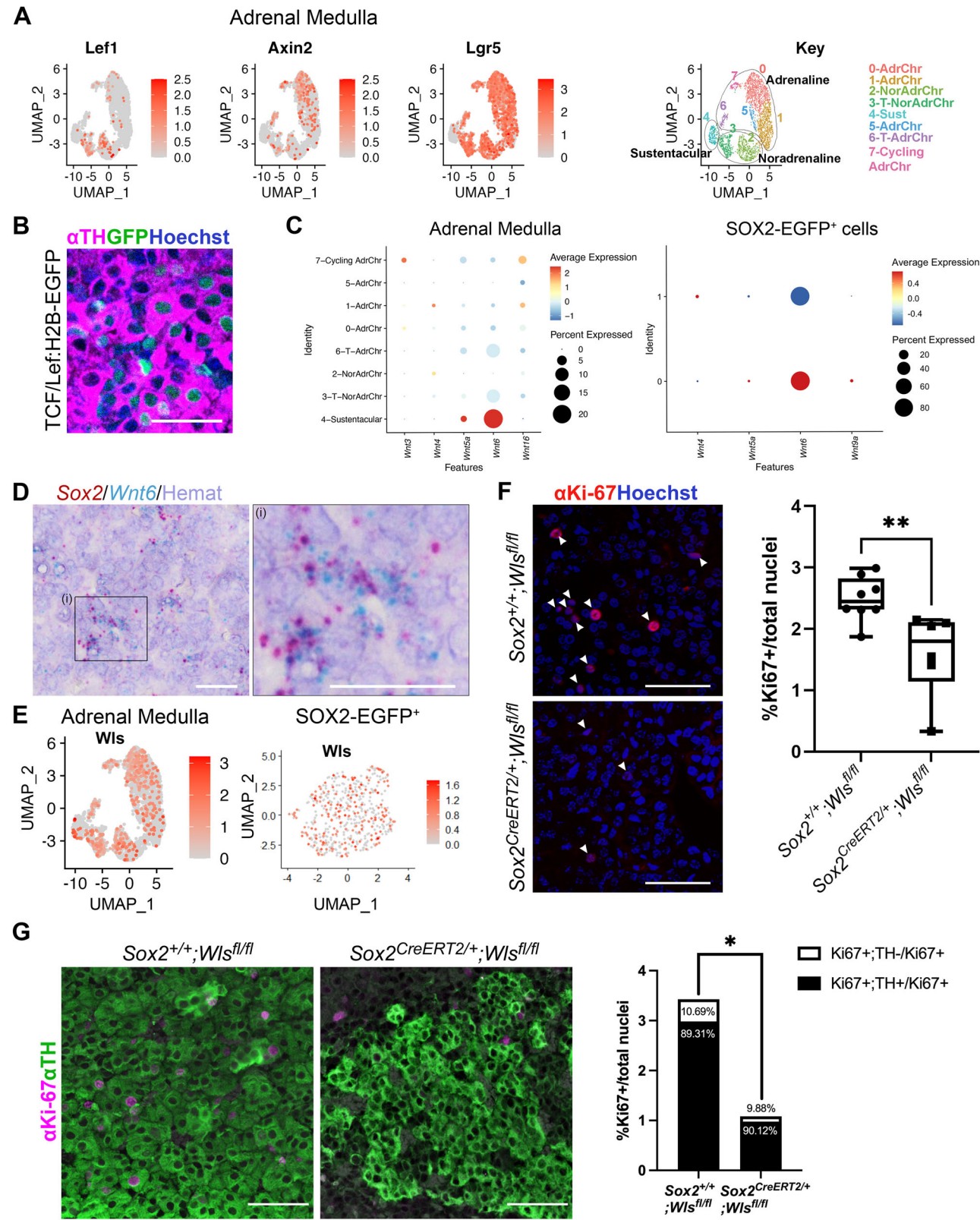

PNMT antibodies were amplified with biotin-streptavidin by incubating at room temperature for 1 h with anti-mouse biotinylated secondary antibody (dilution 1:300, listed in Supplementary Table 1) and Hoechst (Life Technologies, H3570) (dilution 1:10,000), washed 3 times with PBST and incubated at room temperature for 1 h with fluorescent-labelled streptavidin (dilution 1:500, listed in Supplementary Table 1). After washing in PBST, slides were mounted with

Vectashield Antifade Mounting Medium (Vector Laboratories, H-1000-10).

**Immunofluorescence on cryosections**
Sections were blocked for 1 h at room temperature in Blocking Buffer (1% BSA, 0.1% Triton X-100, 5% goat serum). Primary antibodies were diluted in Blocking Buffer at the concentrations reported in

**Fig. 5 | SOX2⁺ adrenomedullary stem cells promote proliferation of chromaffin cells through secretion of paracrine WNT ligands. A** Featureplots for expression of WNT targets *Lef1*, *Axin2* and *Lgr5* in the mouse postnatal adrenal medulla dataset, Key of clusters, grouping by lineage. Colour scale represents Log-normalised expression level, with 0 (grey) representing no expression, and 2.5 (red) for *Lef1* and *Axin2*, or 3 (red) for *Lgr5* representing highest expression. **B** Immunofluorescence using antibodies against TH (chromaffin cells) and GFP (cells that have responded to WNT) on mouse TCF/Lef:H2B-EGFP adrenal medulla at P21. Nuclei are counterstained with Hoechst, scale bars 50 μm. *n* = 1 female, 1 male, 1 technical replicate. **C** Dot plots of all *Wnt* genes expressed in the mouse adrenal medulla dataset (left) and in the isolated SOX2-EGFP⁺ cell dataset (right), subdivided by cell clusters. Colour scale represents average expression level, as a sliding scale of expression magnitude from blue (lowest expression) to red (highest expression). Dot size reflects percentage of cells with gene expression. **D** RNAscope mRNA in situ hybridisation using probes against *Sox2* (red) and *Wnt6* (blue), showing co-expression. Image in right is magnified region (i). Nuclei counterstained with hematoxylin. Scale bars 20 μm. *n* = 2 females, 2 males, 2 technical replicates. **E** Featureplots for *Wls* in the mouse adrenal medulla dataset (left) and in the isolated SOX2-EGFP⁺ cell dataset. Colour scale represents Log-normalised expression level, with 0 (grey) representing no expression, and 3 (red) in 'Adrenal Medulla' or 1.6 (red) in 'SOX2-EGFP⁺' representing highest expression. **F** Representative immunofluorescence using antibodies against Ki-67 marking cycling cells in *Sox2⁺/⁺*; *Wls^fl/fl^* (control, top) and *Sox2^CreERT2/+^*;*Wls^fl/fl^* (mutant, bottom) samples following tamoxifen induction at P13/13/15 and analysis at P21 (*n* = 8 controls, 6 mutants). Nuclei counterstained with Hoechst, scale bars 50 μm. Graph showing percentage of Ki-67 positive cells across replicates, revealing a statistically significant reduction in cycling cells in the mutant. Two-sided unpaired *t*-test, *P*-value = 0.0083. Whiskers min to max, the box extends from the 25th to the 75th percentile, line is median. Source data are provided as a Source Data file. **G** Immunofluorescence staining using antibodies against TH (chromaffin cells, green) and Ki-67 (cycling cells, magenta) in *Sox2⁺/⁺*;*Wls^fl/fl^* (control) and *Sox2^CreERT2/+^*;*Wls^fl/fl^* (mutant) samples following tamoxifen induction at P13/14/15 and analysis at P21 (*n* = 3 controls, 3 mutants). Scale bars 50μm. Bar graph showing percentage of Ki-67 positive cells, split into TH positive cells and non-TH positive cells. Two-sided unpaired *t*-test, *P*-value = 0.0320. Source data are provided as a Source Data file.

Supplementary Table 1 and incubated overnight at 4 °C. After washing 3 times with PBS, secondary fluorophore-conjugated antibodies (dilution 1:500, listed in Supplementary Table 1) and Hoechst (Life Technologies, H3570) (dilution 1:10,000) were diluted in in Blocking Buffer and incubated for 1 h at room temperature. After washing 3 times with PBS, slides were mounted with Vectashield Antifade Mounting Medium (Vector Laboratories, H-1000-10).

### RNAscope mRNA in situ hybridisation
RNAscope was performed on paraffin-embedded sections with the RNAscope 2.5 HD Duplex Kit (ACD Bio, 322430) following the manufacturer's protocol, with optimised retrieval time of 12 min and protease time of 30 min. Probes used are listed in the Supplementary Table 1. Sections were counterstained with Hematoxylin QS (Vector Laboratories, H-3404-100) and slides were mounted in VectaMount Permanent Mounting Medium (Vector Laboratories, H-5000-60).

### Primary cell culture
Adrenal glands were dissected, and the medulla isolated manually. Medullae were placed in an enzymatic digestion mix containing 50 μg/ml DNAse I (Sigma, D5025), 10 mg/ml Collagenase II (Worthington, LS004177), 2.5 μg/ml Fungizone (Gibco, 15290026), 0.1X Trypsin-EDTA (Sigma, 59418C) in 1X Hank's Balanced Salt Solution (HBSS) (Gibco, 14025050). Medullae in enzymatic digestion mix were incubated at 37 °C for 10 min, triturated by pipetting up and down and incubated for 5 min at 37 °C, followed by trituration to obtain a single-cell suspension. Enzymes were inactivated by addition of 10 times volume of serum-containing Base Media: DMEM/F-12 (Gibco, 31330-038) + 5% FBS (Merk, F0804) +50 μ/ml Penicillin-Streptomycin (Gibco, 15070063). The cell suspension was centrifuged at 300 × *g* for 5 min at room temperature, washed twice in PBS before being resuspended in Complete Media: Base Media + 20 ng/ml bFGF (R&D Systems, 234-FSE) +50 μg/ml cholera toxin (Sigma, C8052). Two days after isolation, an equal volume of fresh media was added to each plate. Media was fully changed every 2–3 days. For immunostaining, cells were plated on glass coverslips coated with 0.1% gelatine diluted in PBS.

### Colony forming assay
For colony forming assays, adrenals from *Sox2^eGFP/+^* mice were dissected and tissue was dissociated as described for primary cell culture. GFP⁺ and GFP⁻ cells were separated by flow sorting. After sorting, GFP⁺ and GFP⁻ cells were plated at clonal density of 500 cells/well in a 12-well plate. Two days after isolation, an equal volume of medium to the one present in the plate was added. After that, media were changed every 2–3 days.

After 14 days of culture, cells were washed 3 times in PBS and fixed with 10% NBF for 10 min at room temperature. After washing 3 times with PBS, cells were stained for 10 min with Crystal Violet Solution: 0.5% Crystal Violet powder (Sigma, C0775), 20% methanol in distilled water. Excess crystal violet was washed with running tap water and plates dried before colony observation and imaging.

### Chorioallantoic membrane (CAM) assays
Fertilised Shaver Brown eggs were purchased from Medeggs Ltd and placed in an egg incubator set at 37.8 °C/60% humidity. This is considered developmental day 0. On day 4, the CAM was exposed using curved spring scissors and the window sealed with clear tape to prevent contamination and placed back in the incubator. On day 10 of incubation, a silicone ring of 6 mm diameter was placed onto the CAM of each egg. 8 × 10⁵ isolated *Sox2^eGFP^* cells were seeded within the silicone ring. The window was sealed again and the eggs placed in the incubator until graft collection at day 18.

### Immunofluorescence on cells
For immunofluorescence on cells, coverslips were washed twice in PBS and fixed with 4% PFA on ice for 10 min. After washing with PBST, cells were blocked for 1 h at room temperature in Blocking Buffer (0.15% glycine, 2 mg/ml BSA, 0.1% Triton X-100 in PBS) with 10% sheep serum. Primary antibodies were incubated overnight at 4 °C in Blocking Buffer with 1% sheep serum at the concentrations shown in the Supplementary Table 1. After washing with PBST, sections were incubated for 1 h at room temperature in secondary fluorophore-conjugated antibodies, diluted 1:500 (listed in Supplementary Table 1) in Blocking Buffer with 1% serum. After washing with PBST, coverslips were mounted with Vectashield HardSet Antifade Mounting Medium with DAPI (Vector Laboratories, H-1500-10)

### Imaging
Images of immunofluorescence staining were taken with a Leica TCS SP5 or a Zeiss LSM980 confocal microscope, using an HCX Plan-Apochromat CS 20×/0.7 dry objective and an HCX Plan-Apochromat CS 63×/1.3 Glycine objective (both Leica Microsystems), or Zeiss Plan-Apochromat 20×/0.8 dry objective, a Zeiss C-Apochromat 40×/1.2 Water objective and a Zeiss Plan-Apochromat 63x/1.40 Oil objective. Stacks of 0.5 μm/0.7 μm were taken through the entire section thickness. Immunohistochemistry and RNAscope stained sections were scanned with a Nanozoomer-XR Digital slide scanner (Hamamatsu), close-up images were taken with an Olympus BX34F Brightfield microscope using a 40X objective. Cell culture images were taken with an Olympus Phase Contrast microscope using a 4X or 20X objective. Images were processed with Fiji[43] and with Nanozoomer Digital Pathology View. Figures were created in Adobe Illustrator.

## Statistics and reproducibility

The Experimental Design Assistant by the NC3Rs was consulted to determine sample sizes (https://eda.nc3rs.org.uk/). We assume sex-specific differences in post-pubertal adrenals, therefore each experiment has four groups: control male and female, mutant male and female where both genotype and sex are considered factors of interest. We expected the data to fit parametric assumptions. No data were excluded from the analyses. Experiments were replicated to ensure reproducibility. Where possible, findings have also been verified through independent experimental means i.e. using a different approach. The experiments were not randomized. For experiments necessitating counting and quantification, this was carried out independently by two researchers for each experiment. One of the researchers performed the quantification blind, where genotype and identifying labels had been removed. Cell counting was performed manually with Fiji's "Cell Counter" plugin. For mouse samples, a minimum of three sections per sample were counted. For human samples, 5 representative fields were selected at 20X magnification for each sample and counted. Statistical analysis and graphs (except for single-cell RNA sequencing analysis) were produced using GraphPad Prism.

## Reporting summary

Further information on research design is available in the Nature Portfolio Reporting Summary linked to this article.

## Data availability

All single cell RNA sequencing data generated in this study have been deposited in the Gene Expression Omnibus (GEO) database under accession code GSE237125. The postnatal dataset generated in this study has been deposited in the GEO database under accession code GSM7595834. The isolated stem cell dataset generated in this study has been deposited in the GEO database under accession code GSM7595835. The SCP single cell RNA sequencing data used in this study are available in the GEO database under accession code GSE20125. Source data are provided with this paper.

## Code availability

Code is available at: https://github.com/Andoniadou-Lab/adrenal_stemcell [https://doi.org/10.5281/zenodo.13820404].

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

## Acknowledgements

We thank Professor Martin Fassnacht for support with human adrenal tissue, Professor Karen Liu and Dr William Barrell for support with CAM assays, the King's College London Biological Services facilities, the Advanced Cytometry Platform (Flow Core), Guy's and St. Thomas' NHS Trust and the Genomics Research Platform, R&D Department, Guy's and St. Thomas' NHS Trust. We thank Prof. Françoise Helmbacher and Dr Marika Charalambous for critical reading of the manuscript and insightful comments. Funding: Medical Research Council (MR/T012153/1) to CLA, the Paradifference Foundation to CLA and RO, the Deutsche Forschungsgemeinschaft (DFG German Research Foundation) (Project Number 314061271 – TRR 205) to CLA, SRB and CS.

## Author contributions

Conceptualisation, A.S., C.L.A.; Methodology, A.S., Y.K., C.L.A.; Software and Formal Analysis, A.S., T.L.W., V.Y., L.F., B.K., I.A.; Investigation, A.S., Y.K., T.L.W., I.B., M.E.K., L.F., J.P.R., E.J.L.; Writing – Original Draft, A.S., Y.K., C.L.A.; Writing – Review & Editing, A.S., Y.K., T.L.W., I.B., M.E.K., L.F., J.P.R., E.J.L, V.Y., B.K., R.J.O., P.A., S.R.B., C.S., I.A., C.L.A. Funding Acquisition, C.L.A., R.J.O., S.R.B., C.S.; Resources, B.A., I.A.; Supervision, R.J.O., S.R.B., C.S., I.A., C.L.A.

## Competing interests

A.S. and T.L.W. are currently employees of Altos Labs. V.Y. is currently an employee of Bit.bio. I.B. is currently an employee of Novartis. The remaining authors declare no competing interests.
