## [Peer Review File · Nature Communications]

REVIEWER COMMENTS

Reviewer #1 (Remarks to the Author):

In this study, the authors have used single cell transcriptomics to identify the different chromaffin cell populations in the adrenal medulla. Importantly, using multiple lineage tracing systems they identified a stem cell population, the sustentacular stem cells, that give rise to both adrenaline and noradrenaline stem cells. The authors then showed that these stem cells are derived from embryonic Schwann cell precursors and maintained as a postnatal stem cell population in the adrenal medulla. In addition to functioning as adrenal medulla stem cells, they also support chromaffin cell homeostasis by secreting a Wnt ligand, Wnt 6. In overall, this is an interesting, comprehensive and well-done study that deserves publication, provided that the following issues can be addressed:

Main points:

1. The data presented in Figure 2 are intriguing, but the lineage tracing experiments need to be quantified: Figure 2F and G: % of Sox2 / GFP-positive cells. How many Sox2-cells are neural crest-derived as opposed to SCP-derived?
2. Figure 3E-G is rather weak as presented: it addresses the developmental potential of a cell in a context which has not much to do with the in vivo situation. Nonetheless, addressing the potential of a cell is valid as such, but to do so the data shown in Figure 3G need to be quantified and the numbers have to be compared to the numbers at d0: the few TH / PNMT-positive cells shown in Figure 3G might have been present already at d0; if so, there would be little evidence for a differentiation potential of Sox2-pos. cells in this assay.
3. Likewise, the data in Figure 4D need to be quantified: how many cells differentiated, how many cells maintained stem cell features (Sox2-pos./differentiation marker negative)? The latter would indicate whether there are indeed stem cells, i.e. cells with self-renewal potential.
4. Figure S1J shows pseudotime analysis results from Monocle but the UMAP projection is not matching the one that has been used along the paper for the specific populations. It would be advisable to keep consistency in the representation and to project the pseudotime results on the original UMAP so that the readers can associate the previously defined populations with the

specific pseudotime. In the same subpanel, the marker expression (right side) shows gray color in the plots but the color is not part of the scale and it is not indicated in the figure legend what that color indicates. Please clarify. Moreover, to strengthen the data the authors could use RNAscope to support the existence of the transitioning cell clusters (co-labelling of marker for both cell types).

5. The statement that the Sox2+ sustentacular cells are different from embryonic SCPs needs further evidence. The analysis shown in Figure S3B and C is not showing the differentially expressed genes between the two cell populations. It would be nice to validate these differentially expressed genes e.g. by RNAscope or IF. Moreover, from the Dotplots it seems that the SOX2-enriched transcriptional profile is associated with some redundant GO terms (probably the very same genes included in the “distinct” terms), very few genes are apparently included in each GO term. Despite the fact that the terms seem to be statistically significantly enriched, it is questionable/not clear how much relevance the results have considering the maximum number of genes in a GO term is just 5.

6. In Figure 2E, it is hard to see that Sox10 precedes Sox2 in terms of expression. It might be better to use a different representation, for example, plotting the single features along the pseudotime in the region of interest from the trajectory. Moreover, Figure 2E does not indicate what the color code of the lower subpanel plots represents, and even though it might be intuitive it would be advisable to include the explanation in the figure legend.

7. What is the identity of the proliferating cells shown in Figure 5F? Are these indeed Sox2-negative cells (which would support a paracrine, rather than autocrine effect of Wnt signalling)?

Minor points:

1. The authors state that the adrenal medulla cells are all derived from Schwann cell precursors, however previous studies such as Furlan et al., Science 2017, have shown that neural crest stem cells also likely contribute to adrenal medulla cells (see also main point 1). This should be clarified in the manuscript.

2. Could the authors clarify the sentence in paragraph 88-90? The relation between disease heritability and susceptibility to oncogenic mutations is unclear. In reference 20 for example, it is shown that Sox2+ progenitor cells in the pituitary gland are more prone to oncogenic mutations,

however the tumour mass does not form from these cells suggesting that they may support oncogenesis by paracrine signalling.

3. The new marker for noradrenaline chromaffin cells, PENK, was previously used as a general marker for chromaffin cells in the human adrenal gland, see reference 15. This should be mentioned in the manuscript.

4. Sox10 is not only a marker for stem/progenitor Schwann cells, but also labels the mature glia cell population. Is the idea that by removing the cortex the innervating nerves were removed and hence Sox10 can be used to label only the Schwann cell precursor derived Sox2+ sustentacular cells? This should be commented on.

5. The RNAscope images shown in Figure 2D and Figure 5D are taken at a very low magnification. The authors should provide higher magnifications of these images so that the viewer can see the punctuated dot signal from the transcript.

6. Along the same line, it seems that many Sox2-pos. cells are negative for the markers stained in red in Figure 2D. Is this the case and, if so, what kind of cells might these be? This is relevant given that the authors later perform Sox2-CreERT2 tracing.

7. Lane 130, “expression of markers of previously described sustentacular cells”: please provide reference.

8. In Figure 2C, GFP expression is not evident in the S100B/GFP panel.

9. Lane 239, typo “RNA sequencing”.

10. Figure 5E, the UMAP projection for Wnt6 should also be shown (in the context of Figure 5C).

11. Line 462: indicate Monocle version.

12. Line 479 (Methods): mention the statistical criteria for defining DEGs (p adjusted value thresholds, size of the gene sets analyzed, log2FC thresholds if any were applied...)

13. Line 481: “ClusterProfiler was used to obtain significantly differentially expressed gene ontologies.” The significance threshold used (p adjusted value and/or q value) for the analysis are not indicated.

14. Line 492: add reference to scFates package and version.

Reviewer #2 (Remarks to the Author):

Santambrogio et al. investigate the understudied dynamics of the postnatal adrenal medulla and discover a new stem cell population that replenishes chromaffin cells during growth and homeostasis. While most research has investigated the dynamics of the adrenal cortex, here the authors have optimized a cell isolation protocol to capture viable populations of adrenomedullary cells. In doing so, they not only have defined a chromaffin stem cell population, but also further characterize their transitional states and subpopulations of chromaffin cells in the mouse medulla. In *ova* and *in vivo* mouse work confirm the stem cell nature of these newly defined SOX2+ cells, and computational analysis and conditional deletion studies indicate a need for paracrine signaling from these stem cells to the niche, via WNT6. This is a well-written, elegantly presented study that will be very valuable to the field and future studies. My comments and suggestions for authors are as follows:

Major Comments

1. The authors thoroughly analyzed P15 adrenal glands by single cell RNA-sequencing. While they briefly reference this is a “rapid growth phase”, a broader introduction of what is known of the cell-dynamics of this time-point is suggested. At this time, it is not fully clear as to why they chose this stage.

2. The authors pool sexes for their P15 adrenal medulla analysis, and do not mention any possibility of sexual dimorphism between cell populations and/or signature transcriptomes. Can the single cell sequencing data be separated by sex and any differences confirmed? This would be valuable to include, even if added to supplementary.

3. The SOX2+ cell quantification over time is very strong, and inclusion of human tissue is excellent. The authors have not described in detail as to why they hypothesize an increase occurs between

P15 and P17. A description of the cell dynamics at this postnatal age (see comment 1) is advised to understand this dynamic.

4. In Figure 2C, it is difficult to appreciate the colocalization of SOX2eGFP and S100B. Single panel images are advised to be included here (similar to what is shown in Fig S2C).

5. No color legend has been included in Figure 2E. It is assumed that red indicates positive expression, but what the deep blue coloring is representing in the Sox10 expression plot is not clear. No blue is seen in Sox2 or ChC plot.

6. Single staining of TH and PNMT is not strong validation of chromaffin differentiation in ex vivo explants. Further confirmation that these cells are mouse-derived (e.g. TH can label sympathetic neurons of the chick) and are bona fide chromaffin cells is required. Double staining of known markers and/or use of mouse-specific RNAscope probes for these markers is suggested.

7. The use of the in vivo conditional deletion of Wingless from SOX2 cells is strong showing the need for WNT6 secretion from these cells to influence proliferation of the niche, yet it has not been scrutinized in-depth as to what cells they are influencing. While cycling cells were captured in the RNA-sequencing data, there is no evidence shown to validate this or if other cells do proliferate (it is assumed that as stem cells, SOX2+ cells should also undergo proliferation). Confirmation and quantification of Ki67 with chromaffin and sustentacular cell makers is advised in both wildtype and Wls deleted medulla. The use of EdU/BrdU incorporation is further suggested to provide a more functional marker of active cell cycling.

Minor Comments

1. There is a referencing error on line 166 where the reference has not been included in correct referencing style.

2. Typo in line 239 in word “sequencing”.

3. Referencing errors have occurred in many areas of the Star methods section.

Reviewer #3 (Remarks to the Author):

The study by Santambrogio et al. described a subset of sustentacular cells, marked by SOX2 expression, in the mouse postnatal adrenal medulla, may serve as stem cells for generating all lineages of chromaffin cells. The authors used mouse reporter line to confirm the SOX2 expression in subset of SOX10+ neural crest-derived progenitors. These SOX2+ cells indeed displayed self-renewal property when cultured in vitro and can be further differentiated into chromaffin cells ex vivo. With Sox2 lineage tracing, the cells once expressed SOX2 were highlighted in the adulthood, showing differentiated markers. The authors further probed the Wnt expression in the postnatal adrenal medulla and indicated that SOX2+ cells expressed Wnt6, while the rest of other differentiated lineages all expressed Lgr5. Thereby, the author further concluded that SOX2+ cells promote cell proliferation by paracrine secretion. It is known that sustentacular cells with progenitor characteristics (NESTIN+; partially SOX10+) can proliferate and differentiate into chromaffin and neural lineages. But it is indeed unclear which exact cells are stem cells (self-renewal without fate acquired). This study claims that a group of SOX2+ cells fulfil stem cell function. However, the evidences so far do not conclusively support the claim:

1. Given the fact that the postnatal medulla strongly resembles the embryonic medulla, it is unclear about the identity of the postnatal SOX2+ cells, which is clearly a subset of SOX10+ cells. Are SOX2+/SOX10+ cells just intermediate bridge cells or late Schwann cell precursors (SCPs) before chromaffin differentiation during the postnatal stage? The first scRNA-seq study (10.1126/science.aal3753) describing SCPs has shown the transient expression of Sox2 during the SCP-chromaffin transition. Can authors provide additional evidences to prove that this developmental transition is not happening during the postnatal state?

2. Next, are SOX2+ cells in postnatal gland truly different from the SOX10+ embryonic multipotent SCPs? In the manuscript, the authors compare the DGEs of SOX2+ cells with SCPs dataset from others, indicating that a few GOs are more enriched in SOX2+ cells. It is hard to interpret such vague comparison. I suggest the authors also look into the SOX10+/SOX2- cells versus the SOX10+/SOX2+ cells at postnatal stage, which can potentially provide more evidences of the unique features of this SOX2+ stem cell population.

3. In Fig. 2E, the authors showed the expression levels of Sox10 and Sox2 in the embryonic development trajectory of SCPs. Given the sparse and low expression levels of Sox2, a better data presentation is needed to support the claim "Sox10 expression precedes Sox2 expression". A previous study in human embryonic development of SCPs indicated that SOX10 and SOX2 both show specific TF regulon activity in SCPs. Can authors include the comparison of TF regulons between Sox10 and Sox2 in postnatal dataset in Fig. S2G? It would help to interpret how Sox2 uniquely regulate stemness signalling at this stage.

4. In a previous paper of scRNA-seq of postnatal human and mouse adrenal glands (10.1038/s41467-021-24870-7), a unique human postnatal progenitor cluster was reported, which present a unique set of marker expression (SOX6, NTRK2...) and especially negative in SCP markers (SOX10, S100B...). This subset is not present in postnatal mouse. The authors should compare the SOX2+ stem cells shown here with the previous study. Are the human postnatal progenitor cluster also SOX2 expressing? The authors indicated that this previous study used limited viable medulla cells for scRNA-seq. But it should still be feasible to compare gene marker expression between two cell types. In the mouse dataset of the same study, postnatal SOX10+ cells were described as “glial cells”, which is closely clustered with SCP E13. Can authors look into whether their glial cells also showed SOX2 related expression features?

5. The study also performed a long-term lineage tracing of Sox2+ cells. In this experiment, the cells once expressed SOX2 at postnatal stage were further highlighted in the adulthood. It is not clear the expansion of committed SOX2 progeny is due to the differentiation of SOX2+ cells at adult stage, or the proliferation of early-committed chromaffin cells at postnatal stage. The authors should include the staining of SOX2+ stem cells along with the other differentiated cells among SOX2 progeny. It is also worth trying to do an adult-stage induction, and see whether differentiation would happen at adult stage.

6. It is very interesting that, in the postnatal mouse adrenal gland, the differentiated chromaffin cells express LGR5, but the progenitor cells do not. This has also been shown in previous study. But in human, it is the other way. LGR5 modulates Wnt signalling and usually marks stem cells. Can authors provide more discussions on this? And how this Wnt6-Lgr5 expression pattern may affect the proliferation dynamics in mouse adrenal gland compared to human?

REVIEWER COMMENTS

Reviewer #1 (Remarks to the Author):

In this study, the authors have used single cell transcriptomics to identify the different chromaffin cell populations in the adrenal medulla. Importantly, using multiple lineage tracing systems they identified a stem cell population, the sustentacular stem cells, that give rise to both adrenaline and noradrenaline stem cells. The authors then showed that these stem cells are derived from embryonic Schwann cell precursors and maintained as a postnatal stem cell population in the adrenal medulla. In addition to functioning as adrenal medulla stem cells, they also support chromaffin cell homeostasis by secreting a Wnt ligand, Wnt 6. In overall, this is an interesting, comprehensive and well-done study that deserves publication, provided that the following issues can be addressed:

Main points:

1. The data presented in Figure 2 are intriguing, but the lineage tracing experiments need to be quantified: Figure 2F and G: % of Sox2 / GFP-positive cells. How many Sox2-cells are neural crest-derived as opposed to SCP-derived?

Thank you to the reviewer for asking for this clarification. We have quantified the lineage tracing experiments in Figures 2F and G and included the quantification in the text on page 7. We confirm that 100% of SOX2+ cells originate from Wnt1-Cre expressing cells. Since Sox10-iCre is an inducible driver, we do not expect total recombination, therefore negative cells would not mean that they are not SCP-derived. Despite this, when inducing Sox10iCreERT2/+;R26mTmG/+ at E11.5, we find that only 0.42% of SOX2+ cells are not GFP+ i.e. derive from non-induced cells. Collectively our data indicate that SOX2+ cells are specified from SCPs on or after E11.5 and as we know, SCPs are all neural crest-derived (Kastriti et al. 2022).

2. Figure 3E-G is rather weak as presented: it addresses the developmental potential of a cell in a context which has not much to do with the in vivo situation. Nonetheless, addressing the potential of a cell is valid as such, but to do so the data shown in Figure 3G need to be quantified and the numbers have to be compared to the numbers at d0: the few TH / PNMT-positive cells shown in Figure 3G might have been present already at d0; if so, there would be little evidence for a differentiation potential of Sox2-pos. cells in this assay.

Thank you for this suggestion. We have quantified the data in Figure 3G and included these as a graph. In CAM-derived tissues, SOX2+ cells generate TH+ cells at a rate of 8.51% and

PNMT+ cells at a rate of 5.81% (n=3 CAMs). In contrast, on day zero, we have 0.22% of cultured *Sox2-EGFP* cells being GFP negative (graph in 3D), demonstrating the stark increase in the differentiated cells following *in ovo* culture. We have incorporated these data in Figure 3G and text on page 9. We have also repeated these assays with RNAscope mRNA *in situ* hybridisation (see major comment 6 to reviewer 2) using mouse-specific probes, confirming the neuroendocrine cells are derived from the transplanted cells.

3. Likewise, the data in Figure 4D need to be quantified: how many cells differentiated, how many cells maintained stem cell features (*Sox2*-pos./differentiation marker negative)? The latter would indicate whether there are indeed stem cells, i.e. cells with self-renewal potential.

Quantification of SOX2-derivatives after 1 year, reveals that 8.4% remain SOX2+;GFP+. Quantification of the data in Figure 4E (previously 4D) show 12.5% differentiate into TH+;GFP+, and separate staining for adrenaline and noradrenaline chromaffin cells reveals 9.2% PNMT+;GFP+ and 2.8%PENK+;GFP+, respectively, data on page 9). Furthermore, in response to reviewer 3, we have also included immunofluorescence staining for SOX2 and GFP after one year (new 4D) and included the new 1 year datapoint in the graph (Figure 4C, n=4 *Sox2^{CreERT/+};R26^{mTmG/+}*), demonstrating further expansion of GFP+ clones. Accordingly, we replaced the long-term lineage tracing panel in 4B to a representative one-year tracing. We have adjusted the text on page 9 accordingly.

4. Figure S1J shows pseudotime analysis results from Monocle but the UMAP projection is not matching the one that has been used along the paper for the specific populations. It would be advisable to keep consistency in the representation and to project the pseudotime results on the original UMAP so that the readers can associate the previously defined populations with the specific pseudotime. In the same subpanel, the marker expression (right side) shows gray color in the plots but the color is not part of the scale and it is not indicated in the figure legend what that color indicates. Please clarify. Moreover, to strengthen the data the authors could use RNAscope to support the existence of the transitioning cell clusters (co-labelling of marker for both cell types).

Thank you for pointing this out, we have corrected the Monocle pseudotime UMAP projection to match the one used all along the paper for the specific populations.

As suggested, we have carried out double RNAscope mRNA *in situ* hybridisation at P14 for *Pnmt* and *Sox10* as well as *Penk* and *Sox10*. Although rare, we identify double positive cells. These data are included in the text on page 6 and as new panel S1K.

5. The statement that the *Sox2*+ sustentacular cells are different from embryonic SCPs needs further evidence. The analysis shown in Figure S3B and C is not showing the differentially expressed genes between the two cell populations. It would be nice to validate these differentially expressed genes e.g. by RNAscope or IF. Moreover, from the Dotplots it seems that the SOX2-enriched transcriptional profile is associated with some redundant GO terms (probably the very same genes included in the “distinct” terms), very few genes are apparently included in each GO term. Despite the fact that the terms seem to be statistically significantly

enriched, it is questionable/not clear how much relevance the results have considering the maximum number of genes in a GO term is just 5.

Thank you to the Reviewer; because of this comment we reassessed our approach to the analysis. We agree that the relevance of the results presented in S3B and C can be improved. Therefore we have removed panels S3B and S3C from the manuscript. These are now replaced with a different approach, as follows:

By using the CONOS package to align the SOX2+ postnatal dataset with the developmental glial data and project the developmental cell identities on our postnatal SOX2+ dataset, we observe that the SOX2+ sustentacular cells are not similar to Schwann cell precursors. In fact, they are more similar to differentiated Schwann cells, in particular, postnatal non-myelinating Schwann cells i.e. dissimilar to SCPs (new figure S3B). The top 50 differentially expressed genes from SOX2+ sustentacular cells are scored on both datasets, depicted in new figure S3C. Using SCENIC, we compared the *Sox2* regulons in both of these populations. In developmental glia, there are 121 distinct *Sox2* targets, in the postnatal medulla dataset there are 162 distinct targets, and there are only 7 targets in common (Figure S3D, Supplementary Table 2). The data are described on pages 7 and 8. We therefore consider the SOX2+ sustentacular population to be distinct from embryonic SCPs.

6. In Figure 2E, it is hard to see that *Sox10* precedes *Sox2* in terms of expression. It might be better to use a different representation, for example, plotting the single features along the pseudotime in the region of interest from the trajectory. Moreover, Figure 2E does not indicate what the color code of the lower subpanel plots represents, and even though it might be intuitive it would be advisable to include the explanation in the figure legend.

We have replaced Figure 2E subpanels for *Sox10*, *Sox2* and *Chga* for consistency and included the colour code. In addition, we have added a new panel of single features for *Sox10*, *Sox2* and *Chga*, along pseudotime in the region of interest from the trajectory, as suggested, making it easier to see that *Sox10* expression precedes that of *Sox2*. We have clarified these data in the figure legend.

7. What is the identity of the proliferating cells shown in Figure 5F? Are these indeed *Sox2*-negative cells (which would support a paracrine, rather than autocrine effect of Wnt signalling)?

The scRNAseq data in Figure 1B-E reveal that the proliferating cells mostly express chromaffin markers. We have added a new Figure 5G, showing double immunofluorescence staining for Ki-67 and TH, and a graph showing the breakdown of the cycling cells. From this it is clear that the greatest reduction in cycling cells is attributed to TH+ chromaffin cells. Additionally, in a new supplementary panel S5C, we showing RNAscope mRNA *in situ* hybridisation, using probes against *Sox2* and *Mki67*, or *Th* and *Mki67* on control (as well as *Sox2*^{CreERT2/+}; *Wls*^{fl/fl} samples). These analyses demonstrate that normally, cycling cells are found mostly outside of the *Sox2*-expressing compartment (majority being *Th*-expressing), with rare *Sox2*-expressing cycling cells. Therefore, the data support a paracrine effect of WNT signalling. We cannot exclude an autocrine effect as well, but the number of double positive *Sox2*+;*Mki67*+ cells is too low to determine this.

Minor points:

1. The authors state that the adrenal medulla cells are all derived from Schwann cell precursors, however previous studies such as Furlan et al., Science 2017, have shown that neural crest stem cells also likely contribute to adrenal medulla cells (see also main point 1). This should be clarified in the manuscript.

We have clarified this on page 3 by explicitly stating that SCPs are neural crest-derived.

2. Could the authors clarify the sentence in paragraph 88-90? The relation between disease heritability and susceptibility to oncogenic mutations is unclear. In reference 20 for example, it is shown that Sox2+ progenitor cells in the pituitary gland are more prone to oncogenic mutations, however the tumour mass does not form from these cells suggesting that they may support oncogenesis by paracrine signalling.

Apologies for any confusion, we were not trying to relate disease heritability and susceptibility to oncogenic mutations, rather to present the population as a good target of oncogenic mutations for generating models of disease. We have changed the paragraph on page 3 to the following:

'The identification of this novel adrenomedullary stem cell population holds promise for applications in regenerative medicine in neuroendocrine structures. Pheochromocytomas and paragangliomas, which present some of the highest rates of gene heritability across all tumours, are understudied due to lack of genetic tools. This adrenomedullary stem cell population constitutes an ideal target for oncogenic mutations, in the quest for the generation of novel animal models'.

3. The new marker for noradrenaline chromaffin cells, PENK, was previously used as a general marker for chromaffin cells in the human adrenal gland, see reference 15. This should be mentioned in the manuscript.

We have now included that PENK has been confirmed as a general marker of fetal human chromaffin cells on page 4 and referenced this appropriately.

4. Sox10 is not only a marker for stem/progenitor Schwann cells, but also labels the mature glia cell population. Is the idea that by removing the cortex the innervating nerves were removed and hence Sox10 can be used to label only the Schwann cell precursor derived Sox2+ sustentacular cells? This should be commented on.

In Furlan 2017, the embryonic neural-crest derived SCPs are referred to as multipotent peripheral glial cells. If we have understood correctly, the Reviewer is referring to glial cells in the postnatal adrenal. These will be the same as the sustentacular cells, a proportion of which express GFAP and SOX10. The innervating nerves of the cortex have indeed been removed, and the population of SOX10+ cells we are investigating are only SCP-derived. We have explicitly stated the removal of innervating nerves in the methods on page 19.

5. The RNAscope images shown in Figure 2D and Figure 5D are taken at a very low magnification. The authors should provide higher magnifications of these images so that the viewer can see the punctuated dot signal from the transcript.

We have provided higher magnification images for Figure 2D. The boxed area in Figure 5D is shown at higher magnification.

6. Along the same line, it seems that many Sox2-pos. cells are negative for the markers stained in red in Figure 2D. Is this the case and, if so, what kind of cells might these be? This is relevant given that the authors later perform Sox2-CreERT2 tracing.

We have clarified that Sox2+ cells include subsets of *S100b*+, *Gfap*+, *Sox10*+ and *Plp1*+ cells and provide a Venn diagram showing the overlap (Supplementary Figure S2C and referred to on page 7). The vast majority of the sustentacular cells co-express two or more markers among *S100b*, *Gfap*, *Sox10*, *Plp1* and *Sox2*.

7. Lane 130, “expression of markers of previously described sustentacular cells”: please provide reference.

We have provided four references for this statement: Cocchia & Michetti, 1981; Magro & Grasso 1997; Suzuki & Kachi, 1995; Kastriti et al. 2020. This is now on line 133.

8. In Figure 2C, GFP expression is not evident in the S100B/GFP panel.

We have split the channels in Figure 2C to show the individual panels, making the GFP expression easier to appreciate.

9. Lane 239, typo “RNA sequencing”.

Thank you, we have corrected this.

10. Figure 5E, the UMAP projection for *Wnt6* should also be shown (in the context of Figure 5C).

Thank you, we include these plots for *Wnt6* in Supplementary 5A and Supplementary 5B and reference these figures in the text on page 10.

11. Line 462: indicate Monocle version.

We have included the Monocle version.

12. Line 479 (Methods): mention the statistical criteria for defining DEGs (p adjusted value thresholds, size of the gene sets analyzed, log2FC thresholds if any were applied...)

We have included that we applied a log2FC threshold of 0.25. In the methods we indicated that we excluded cells with less than 1000 genes and more than 5000 genes. Therefore we are analysing an average of 3000 genes per cell.

13. Line 481: "ClusterProfiler was used to obtain significantly differentially expressed gene ontologies." The significance threshold used (p adjusted value and/or q value) for the analysis are not indicated.

We have removed this analysis in response to comment 5.

14. Line 492: add reference to scFates package and version.

Thank you, we have included this.

Reviewer #2 (Remarks to the Author):

Santambrogio et al. investigate the understudied dynamics of the postnatal adrenal medulla and discover a new stem cell population that replenishes chromaffin cells during growth and homeostasis. While most research has investigated the dynamics of the adrenal cortex, here the authors have optimized a cell isolation protocol to capture viable populations of adrenomedullary cells. In doing so, they not only have defined a chromaffin stem cell population, but also further characterize their transitional states and subpopulations of chromaffin cells in the mouse medulla. In ova and in vivo mouse work confirm the stem cell nature of these newly defined SOX2+ cells, and computational analysis and conditional deletion studies indicate a need for paracrine signaling from these stem cells to the niche, via WNT6. This is a well-written, elegantly presented study that will be very valuable to the field and future studies. My comments and suggestions for authors are as follows:

Major Comments

1. The authors thoroughly analyzed P15 adrenal glands by single cell RNA-sequencing. While they briefly reference this is a "rapid growth phase", a broader introduction of what is known of the cell-dynamics of this time-point is suggested. At this time, it is not fully clear as to why they chose this stage.

We have clarified the premise of choosing this age. We have revised the sentences on lines 97-100 to: "To investigate the postnatal cell composition of the adrenal medulla, we performed droplet-based single-cell RNA sequencing on 10 mouse adrenals that were manually dissected to remove the majority of the cortex, at postnatal day (P) 15. At this time of rapid postnatal growth (reference 17), we seek to capture a plastic state that is still representative of normal homeostasis of the organ."

2. The authors pool sexes for their P15 adrenal medulla analysis, and do not mention any possibility of sexual dimorphism between cell populations and/or signature transcriptomes. Can the single cell sequencing data be separated by sex and any differences confirmed? This would be valuable to include, even if added to supplementary.

We thank the reviewer for this valuable comment, we have separated the data by sex and explored differences. As already shown in figure S1D via expression of *Xist*, we do not see

discrepancies in cell composition. We performed analysis of differentially expressed genes, and report the findings on page 6 of the manuscript and as a heatmap in S1D.

To summarise these findings, other than expected differences in expression of *Xist* and *Tsix*, as well as Y-linked *Ddx3y* and *Eif2s3y*, we observe only a handful of differences across the different clusters. Female adrenaline chromaffin cells show an increase of *Scg2*, female noradrenaline chromaffin cells have increased expression of *Rnd2*, and male transitioning noradrenaline cells have an increase of two major histocompatibility complex genes *B2m* and *H2-D1*. Taking these data together with the lack of sex-specific differences in lineage tracing from *Sox2*-expressing cells, we conclude that there is no obvious sexual dimorphism at this stage.

3. The SOX2+ cell quantification over time is very strong, and inclusion of human tissue is excellent. The authors have not described in detail as to why they hypothesize an increase occurs between P15 and P17. A description of the cell dynamics at this postnatal age (see comment 1) is advised to understand this dynamic.

To address this comment, we have analysed cycling cells expressing Ki-67 from P15. We find an increase in cycling activity from P15 to P17, correlating with the increase in SOX2+ cells observed in Figure 2B, suggesting that it might be the SOX2+ stem cells that undergo a burst of self-renewal at this stage. We included these data as a new addition to Supplementary Figure S2A and referred to them on lines 158-159. To our knowledge, this is the first report touching upon cell dynamics of the adrenal medulla at these stages but we do not have a hypothesis relating to the physiological function of this expansion.

4. In Figure 2C, it is difficult to appreciate the colocalization of SOX2eGFP and S100B. Single panel images are advised to be included here (similar to what is shown in Fig S2C).

We have now shown the individual channels, also requested by Reviewer 1, minor point 8.

5. No color legend has been included in Figure 2E. It is assumed that red indicates positive expression, but what the deep blue coloring is representing in the Sox10 expression plot is not clear. No blue is seen in Sox2 or ChC plot.

We have replaced Figure E subpanels for *Sox10*, *Sox2* and *Chga* for consistency, and included the colour code. In addition, we have added a new panel of single features for *Sox10*, *Sox2* and *Chga*, along pseudotime in the region of interest from the trajectory, as suggested. We have clarified these data in the figure legend. These were also requested by Reviewer 1, main point 6.

6. Single staining of TH and PNMT is not strong validation of chromaffin differentiation in vivo explants. Further confirmation that these cells are mouse-derived (e.g. TH can label sympathetic neurons of the chick) and are bona fide chromaffin cells is required. Double

staining of known markers and/or use of mouse-specific RNAscope probes for these markers is suggested.

We have repeated these assays with RNAscope mRNA *in situ* hybridisation using mouse-specific probes against *Sox2* and *Th*, confirming the neuroendocrine cells are derived from the transplanted cells. We have incorporated these data as a new panel in Figure 3G and text on page 229-232.

7. The use of the *in vivo* conditional deletion of Wingless from SOX2 cells is strong showing the need for WNT6 secretion from these cells to influence proliferation of the niche, yet it has not been scrutinized in-depth as to what cells they are influencing. While cycling cells were captured in the RNA-sequencing data, there is no evidence shown to validate this or if other cells do proliferate (it is assumed that as stem cells, SOX2+ cells should also undergo proliferation). Confirmation and quantification of Ki67 with chromaffin and sustentacular cell markers is advised in both wildtype and WIs deleted medulla. The use of EdU/BrdU incorporation is further suggested to provide a more functional marker of active cell cycling.

In our response to Reviewer 1, main point 7, we highlighted that the scRNAseq data in Figure 1B-E show that the proliferating cells mostly express chromaffin markers. We carried out double immunofluorescence staining using antibodies against TH and Ki-67 on control adrenals, demonstrating the majority of the Ki-67+ cells are TH+ chromaffin cells. We included these data in Figure 5G. As advised by the Reviewer, we compared controls to *Sox2^{CreERT2/+};Wls^{fl/fl}* mutants, where there is a significant reduction. As suggested, we have carried out pulse EdU experiments in control *Sox2^{+/+};Wls^{fl/fl}* and *Sox2^{CreERT2/+};Wls^{fl/fl}* mutant mice, which show a clear reduction in incorporation of the thymidine analogue in the mutants. The data are shown in Figure S5D. Taken together, these new data support the paracrine effect of WNT signals originating from the SOX2+ stem cells.

Additionally, the Reviewer points out that SOX2+ cells, as stem cells, should also undergo proliferation. In Figure 1E we show that the sustentacular cell cluster includes cells in G2M and S phase and demonstrate an expanding population through lineage tracing *in vivo*, which strongly support their capacity to proliferate (Figure 4). To specifically demonstrate the existence of cycling *Sox2*+ cells, we carried out RNAscope mRNA *in situ* hybridisation using probes against *Sox2* and *Mki67*. Although the majority of cycling cells are *Sox2* negative (majority are *Th* positive), rare double positive *Sox2/Mki67* cells can be identified (Figure S5C).

Minor Comments

1. There is a referencing error on line 166 where the reference has not been included in correct referencing style.

Thank you, we have corrected this.

2. Typo in line 239 in word "sequencing".

Thank you, we have corrected this.

3. Referencing errors have occurred in many areas of the Star methods section.

We apologise for this oversight, we have gone through the manuscript and corrected errors.

Reviewer #3 (Remarks to the Author):

The study by Santambrogio et al. described a subset of sustentacular cells, marked by SOX2 expression, in the mouse postnatal adrenal medulla, may serve as stem cells for generating all lineages of chromaffin cells. The authors used mouse reporter line to confirm the SOX2 expression in subset of SOX10+ neural crest-derived progenitors. These SOX2+ cells indeed displayed self-renewal property when cultured in vitro and can be further differentiated into chromaffin cells ex vivo. With Sox2 lineage tracing, the cells once expressed SOX2 were highlighted in the adulthood, showing differentiated markers. The authors further probed the Wnt expression in the postnatal adrenal medulla and indicated that SOX2+ cells expressed Wnt6, while the rest of other differentiated lineages all expressed Lgr5. Thereby, the author further concluded that SOX2+ cells promote cell proliferation by paracrine secretion. It is known that sustentacular cells with progenitor characteristics (NESTIN+; partially SOX10+) can proliferate and differentiate into chromaffin and neural lineages. But it is indeed unclear which exact cells are stem cells (self-renewal without fate acquired). This study claims that a group of SOX2+ cells fulfil stem cell function. However, the evidences so far do not conclusively support the claim:

1. Given the fact that the postnatal medulla strongly resembles the embryonic medulla, it is unclear about the identity of the postnatal SOX2+ cells, which is clearly a subset of SOX10+ cells. Are SOX2+/SOX10+ cells just intermediate bridge cells or late Schwann cell precursors (SCPs) before chromaffin differentiation during the postnatal stage? The first scRNA-seq study (10.1126/science.aal3753) describing SCPs has shown the transient expression of Sox2 during the SCP-chromaffin transition. Can authors provide additional evidences to prove that this developmental transition is not happening during the postnatal state?

We thank the Reviewer for bringing this up as it is very interesting, conceptually. In Furlan et al. 2017, Figure S7B, Sox2 is within a list of genes significantly increasing during SCP to chromaffin cell transition. That data do not indicate if this is transient expression or not, but it is supportive that Sox2 expression follows SCP specification, and initiates prior to chromaffin cell differentiation. Perhaps the Reviewer may also have been thinking about Kastriiti et al. 2022, where the authors (also authors on this manuscript), report that the Sox2 regulon is active in SCPs before commitment to terminal fates. The conclusion that activity of these regulons may convey multipotency expression is supportive of SOX2 having an active role in the stem cell state in the postnatal adrenal.

Since adrenal development does not stop at birth, the transition from an SCP to a chromaffin cell can continue. The evidence shown here, regarding the persistence of the population in mouse and humans (Figure 2A,B, Figure S2B), persistence of SOX2+ cells after long-term lineage tracing (Figure 4D) as well as the contribution of new chromaffin cells from adult mice (Figure 4E), all point to a long-lived SOX2+ cell, and not a transient state overall. This does not preclude that some cells going through this transient state co-exist in the adult.

2. Next, are SOX2+ cells in postnatal gland truly different from the SOX10+ embryonic multipotent SCPs? In the manuscript, the authors compare the DGEs of SOX2+ cells with SCPs dataset from others, indicating that a few GOs are more enriched in SOX2+ cells. It is hard to interpret such vague comparison.

Supportive of a distinct cell type, we have included new analyses showing that SOX2+ cells in the postnatal mouse are distinct from SCPs (new Figures S3B and C) and that the SOX2 regulons in these two populations are dissimilar (also see answer to Reviewer 1, point 5). By using the CONOS package to align the SOX2+ postnatal dataset with the developmental glial data and project the developmental cell identities on our postnatal SOX2+ dataset, we observe that the SOX2+ sustentacular cells are dissimilar to Schwann cell precursors. Pinpointing the cell type they are most similar to among neural crest derivatives, we found a closer match to postnatal differentiated non-myelinating Schwann cells instead (new Figure S3B). The top 50 differentially expressed genes from SOX2+ sustentacular cells are scored on both datasets, depicted in new Figure S3C. Using SCENIC, we compared the Sox2 regulons in both of these populations. In developmental glia, there are 121 distinct Sox2 targets, in the postnatal medulla dataset there are 162 distinct targets, and there are only 7 targets in common (Figure S3D, Supplementary Table 2). We therefore consider the SOX2+ sustentacular population to be distinct from embryonic SCPs. We have removed the analysis previously presented in S3B and S3C from the manuscript.

3. In Fig. 2E, the authors showed the expression levels of Sox10 and Sox2 in the embryonic development trajectory of SCPs. Given the sparse and low expression levels of Sox2, a better data presentation is needed to support the claim “Sox10 expression precedes Sox2 expression”. A previous study in human embryonic development of SCPs indicated that SOX10 and SOX2 both show specific TF regulon activity in SCPs. Can authors include the comparison of TF regulons between Sox10 and Sox2 in postnatal dataset in Fig. S2G? It would help to interpret how Sox2 uniquely regulate stemness signalling at this stage.

Thank you for this comment, we have added a new representation of the expression over pseudotime in Figure 2E. Additionally, as requested, we have added the specific TF regulon activity of Sox2 and Sox10 in the postnatal datasets in Figure S2D and Supplementary Table 1, and summarised the data on lines 180-182. The majority of targets are unique to each TF, supporting a distinct role for SOX2 to regulate stemness at this stage (38 targets shared, out of 485 SOX10 targets and 169 SOX2 targets).

4. In a previous paper of scRNA-seq of postnatal human and mouse adrenal glands (10.1038/s41467-021-24870-7), a unique human postnatal progenitor cluster was reported, which present a unique set of marker expression (SOX6, NTRK2...) and especially negative in SCP markers (SOX10, S100B...). This subset is not present in postnatal mouse. The authors should compare the SOX2+ stem cells shown here with the previous study. Are the human postnatal progenitor cluster also SOX2 expressing? The authors indicated that this previous study used limited viable medulla cells for scRNA-seq. But it should still be feasible to compare gene marker expression between two cell types. In the mouse dataset of the same study, postnatal SOX10+ cells were described as “glial cells”, which is closely clustered with SCP E13. Can authors look into whether their glial cells also showed SOX2 related expression features?

In this human study (Bedoya-Reina, *et al.* (2021)), SOX2 is found to be expressed in a small number of cells of the human postnatal progenitor cluster, confirming our *in vivo* IHC data in Figure S2B. We assessed the marker expression of the human cholinergic progenitor population (*Ntrk2*, *ErbB3*, *Sox6*, *Rtnn*) and did not find these to mirror our *Sox2*-expressing stem cells in mouse. *Sox2*-expressing cells are mutually exclusive with *Ntrk2*-expressing cells in the mouse adrenal medulla (Figure 1, below), which are expressed in the adrenaline-producing chromaffin cell clusters. Although *ErbB3* is not exclusive to sustentacular cells, it is expressed by *Sox2*-expressing cells as well. A small subset of *Sox2*-expressing cells express *Sox6*, which is not specific to this population. *Rtnn* shows low, sporadic expression across the medullary populations including *Sox2*-expressing cells.

Although we are not including these analyses in the manuscript, we provide the plots here for the Reviewers:

Figure 1 Expression of cholinergic progenitor markers reported in Bedoya-Reina, *et al.* (2021), in our 'whole medulla' mouse dataset.

Figure 2 Expression of cholinergic progenitor markers reported in Bedoya-Reina, *et al.* (2021), in our 'Sox2-expressing only' mouse medulla dataset.

5. The study also performed a long-term lineage tracing of Sox2+ cells. In this experiment, the cells once expressed SOX2 at postnatal stage were further highlighted in the adulthood. The authors should include the staining of SOX2+ stem cells along with the other differentiated cells among SOX2 progeny. It is also worth trying to do an adult-stage induction, and see whether differentiation would happen at adult stage.

We have added immunostaining using antibodies against SOX2, in lineage-traced samples. These confirm the persisting presence of SOX2+ cells. We have included these data in new Figure 4D. Thank you to the reviewer for suggesting induction at adult stages, this is already shown in existing Figure 4E. This reveals that new chromaffin cells are generated from SOX2+ cells in the adult, with 8.45% of labelled lineage-traced cells being SOX2+ 1 year after induction (roughly 2% of the total cells in the adrenal medulla). As mentioned in response to Reviewer 1, main point 3, we have also included 1 year of lineage tracing as a new datapoint in 4C (graph) and 4B (image), demonstrating further expansion of GFP+ clones.

6. It is very interesting that, in the postnatal mouse adrenal gland, the differentiated chromaffin cells express LGR5, but the progenitor cells do not. This has also been shown in previous study. But in human, it is the other way. LGR5 modulates Wnt signalling and usually marks stem cells. Can authors provide more discussions on this? And how this Wnt6-Lgr5 expression pattern may affect the proliferation dynamics in mouse adrenal gland compared to human?

We are not certain that the equivalent cell type in humans has LGR5 expression, since cholinergic progenitors are not a faithful equivalent cell type to the Sox2-expressing cells in mouse; they do not express SOX2, despite the presence of a SOX2+ population in human (Figure S2B). With additional transcriptomic characterisation of the human adrenal medulla in future studies, to include a larger number of cells, we will be able to draw better parallels. The findings we present here in the mouse adrenal with activation of target genes in committing progenitor and differentiated cell lineages, suggest that the stem cells do not rely on activation of the WNT pathway at the time points analysed, so expression of *Wnt6* from the

stem cells would be primarily acting in a cell non-autonomous (paracrine) fashion, rather than cell-autonomous (autocrine). In fact, the expression of *Lgr5* in the committed cells supports this, since, in addition to being a target, it can enhance the WNT signal response in the presence of Rspodins. This makes sense for an organ where a main source of proliferation is the committed neuroendocrine cells. This is also supported by the composition of the proliferating cell cluster in the dataset presented. It is exciting that the stem cells therefore have a fundamental role in promoting this turnover from committed cells through WNT, in addition to the direct generation of new cells, a process which is unlikely to rely on WNT activation. We have expanded the Discussion on page 12, lines 321-325, to include this point.

REVIEWERS' COMMENTS

Reviewer #1 (Remarks to the Author):

The authors have responded very carefully to all my previous comments, further improving their nice story!

Reviewer #2 (Remarks to the Author):

The authors have addressed all my comments and have provided thorough explanation and ample data in their responses.

Reviewer #3 (Remarks to the Author):

My previous concern regards to the unique stem cell identity of SOX2+ cells apart from SOX10+ cells in postnatal adrenal medulla, and how these cell populations progress across the developmental stages. This comment is also mentioned by Review 1. The authors have answered this question by providing additional bioinformatics analysis, and lineage tracing at adult stage. Additional discussion has also been incorporated on the role of WNT signalling in building the sustentacular cell niche. I have no further comments.